# Increasing the Accuracy of the Characterization of a Thin Semiconductor or Dielectric Film on a Substrate from Only One Quasi-Normal Incidence UV/Vis/NIR Reflectance Spectrum of the Sample

**DOI:** 10.3390/nano13172407

**Published:** 2023-08-24

**Authors:** Dorian Minkov, George Angelov, Emilio Marquez, Rossen Radonov, Rostislav Rusev, Dimitar Nikolov, Susana Ruano

**Affiliations:** 1Scientific Research Section (NIS), Technical University, 1000 Sofia, Bulgaria; 2Department of Microelectronics, Faculty of Electronics Engineering and Technologies, Technical University, 1000 Sofia, Bulgaria; angelov@ecad.tu-sofia.bg (G.A.); rossen.radonov@ecad.tu-sofia.bg (R.R.); rusev@ecad.tu-sofia.bg (R.R.); dnikolov@elsys-bg.org (D.N.); 3Faculty of Science, Department of Condensed-Matter Physics, University of Cadiz, Puerto Real, 11510 Cadiz, Spain; emilio.marquez@uca.es; 4Photovoltaic Solar Energy Unit, Centre for Energy, Environmental and Technological Research (CIEMAT), Avenida Complutense 40, 28040 Madrid, Spain; susanamaria.fernandez@ciemat.es

**Keywords:** thin film, semiconductor or insulator, accurate characterization, reflectance spectrum, optimized envelope method, spectrophotometry, dispersion model

## Abstract

OEMT is an existing optimizing envelope method for thin-film characterization that uses only one transmittance spectrum, *T*(*λ*), of the film deposited on the substrate. OEMT computes the optimized values of the average thickness, $\bar{d}$, and the thickness non-uniformity, Δd, employing variables for the external smoothing of *T*(*λ*), the slit width correction, and the optimized wavelength intervals for the computation of $\bar{d}$ and Δd, and taking into account both the finite size and absorption of the substrate. Our group had achieved record low relative errors, <0.1%, in $\bar{d}$ of thin semiconductor films via OEMT, whereas the high accuracy of $\bar{d}$ and Δd allow for the accurate computation of the complex refractive index, $\dot{N}$(*λ*), of the film. In this paper is a proposed envelope method, named OEMR, for the characterization of thin dielectric or semiconductor films using only one quasi-normal incidence UV/Vis/NIR reflectance spectrum, *R*(*λ*), of the film on the substrate. The features of OEMR are similar to the described above features of OEMT. OEMR and several popular dispersion models are employed for the characterization of two a-Si films, only from *R*(*λ*), with computed $\bar{d}$ = 674.3 nm and Δd = 11.5 nm for the thinner film. It is demonstrated that the most accurate characterizations of these films over the measured spectrum are based on OEMR.

## 1. Introduction

Thin semiconductor and dielectric films are used in photonic circuits, solar cells, thin-film transistors, holography, thin-film batteries, etc. [1,2]. Using such a film in a device requires knowledge of its characteristics, which depends on its composition and preparation technology [3,4]. The optical characteristics of a film can be derived via optical characterization, where a narrow light beam irradiates the surface of the film, thus forming a light spot there [5,6]. The main spectral characteristic of a film, which can be determined using such techniques, is its complex refractive index $\dot{N}\left(\lambda \right)=n\left(\lambda \right)+ik\left(\lambda \right)$, where *n*(*λ*) is the refractive index, *k*(*λ*) is the extinction coefficient, *λ* is the wavelength of light, and “i” is the imaginary unit [7,8]. The main thickness parameters of a film, measurable using such techniques, are the average thickness, $\bar{d}$, over the light spot and the thickness non-uniformity, ∆d = [max(*d*) − min(*d*)]/2 ≥ 0, over the same spot, where *d* is the thickness in a particular point within the spot. Additionally, ∆d can be described as a larger-scale version of the surface roughness of the film [9].

Film characterizations are facilitated by employing dispersion models (DMs), usually based on electron oscillators, which can be classified as narrow-spectrum DMs and broad-spectrum DMs. Narrow-spectrum DMs usually formulate *n*(*λ*) or *k*(*λ*) only in the spectral region of weak and medium absorption, where *n*^2^(*λ*) >> *k*^2^(*λ*) and *E* < *E*_b_ > *E*_g_, where *E* (eV) = 1239.8/*λ* (nm) is the photon energy, *E*_g_ is the optical bandgap, and *E*_b_ is the photon energy above which interband transitions prevail. The Wemple–DiDomenico model, which is of this type, representing an undamped single oscillator, is valid for covalent, ionic, glassy, and amorphous semiconductor materials and provides the following [10,11]:(1)$$\frac{1}{n^{2}-1}≅\frac{E_{0}}{E_{d}}-\frac{E^{2}}{E_{0}E_{d}}\mathrm{for}\;E<E_{b}>E_{g},$$
where *E*_0_ and *E*_d_ are the energy and the strength of the oscillator. Therefore, *E*_0_ and *E*_d_ can be calculated via the linear regression from a Wemple–DiDomenico plot depicting (*n*^2^ − 1)^−1^ as a function of *E*^2^ [11]. It is also seen from Equation (1) that the static refractive index can be expressed as *n*(0) = *n*(*E* = 0) = (1 + *E*_d_/*E*_0_)^1/2^.

In addition, the absorption coefficient *α* = 4π*k*/*λ* of amorphous materials in the region of weak and medium absorption can be approximated using the Urbach rule, corresponding to the structural disorder-generating localized electronic states and the Urbach tail as follows [12,13]:(2)$$\alpha \approx \alpha_{0}\exp\left(\frac{E}{E_{U}}\right)\mathrm{for}\;E<E_{b}>E_{g},$$
where *α*_0_ = *α*(*E*→0) > 0, and the Urbach energy *E*_U_ quantifies the energetic disorder in the band edges. However, Equation (2) is inaccurate for very long wavelengths because it provides *α*(*E*→0) ≈ *α*_0_ > 0, although *α*(*E* = 0) = 4π*k*(0)/*λ*_∞_ ~ 4π*k*(0)*E*(0) = 0.

Broad-spectrum DMs formulate the complex dielectric function $\dot{\varepsilon }\left(E\right)=\varepsilon_{r}\left(E\right)+i\varepsilon_{i}\left(E\right)={\dot{N}}^{2}=n^{2}-k^{2}+i2nk$ or the respective complex refractive index of a material over wider spectra, including the UV/Vis/NIR spectral region with wavelengths *λ* ≈ [150, 3000] nm [14,15]. Arguably the most popular broad-spectrum DMs are the Tauc–Lorentz DM of Jellison and Modine (TL) [16,17,18], the Campi–Coriasso DM (CC) [19,20], and the new amorphous dispersion formula DM (NA) [21,22]. TL uses the Tauc joint density of states and a Lorentz oscillator representing a damped harmonic oscillator. CC is a version of TL, in which the normalized oscillator strength is shifted by *E*_g_, which is achieved by substituting *E* with *E* − *E*_g_. NA has been established in order to give a Lorenzian shape to the expressions for *n*(*E*) and *k*(*E*). The main problem concerning the utilization of TL, CC, and NA for the characterization of amorphous materials is that in all of them, *k*(*λ*) = 0 is assumed for *E* < *E*_g_, thus leading to characterization errors in the range *λ* > *λ*_g_ (nm) = 1239.8/*E*_g_ (eV).

Unlike the above three DMs, the Gaussian DM (GA) provides symmetrical *ε*_i_(*E*), which makes it applicable for descriptions of absorption with peaks at *E* < *E*_g_ associated with the disorder, dangling bonds, impurities, and vacancies [23,24]. Furthermore, the contributions to the dielectric function caused via absorption with peaks above the measured energy spectral range can be included as a pole (PE) in this DM, i.e., a Lorentz oscillator with vanishing broadening [25,26].

With regard to the above two paragraphs, the number of employed oscillators is printed before the two-letter abbreviation of any DM allowing the usage of multiple oscillators henceforth in this paper. The exemplary calculated real and imaginary parts of the complex dielectric function and its respective normal incidence reflectance, *R* = [(*n* − 1)^2^ + *k*^2^]/[(*n* + 1)^2^ + *k*^2^], at the boundary between air and material, are depicted in Figure 1 for 1TL, 1CC, 1NA, and 1GA.

It is inferred from Figure 1a that max[*ε*_r_(*E*)] and max[*ε*_i_(*E*)] are positioned at the highest energies for 1NA as well as that the refractive index *n*{*E* = [0,max(*ε*_r_)]} is the largest also for 1NA. Accordingly, the data from Figure 1c indicate that the reflectance of UV/Vis/NIR light is significantly larger, assuming the validity of NA compared to TL, CC, and GA. It is also apparent from Figure 1 that the results regarding TL are the closest to these for CC.

To increase the accuracy of the characterization of amorphous materials, the broad-spectrum single oscillator Tauc–Lorentz–Urbach DM (TLU) has been developed. There are two versions of TLU, and in both of them is the assumed validity of TL for *E* > *E*_b_ and the Urbach tail for *E* ≤ *E*_b_. The TLU developed by Foldyna (TLUF) has used the expression *ε*_i_(*E*) = const/*E* × exp(*E*/*E*_U_) in the range *E* ≤ *E*_b_, assuming that the Urbach tail is formulated via Equation (2), and the variation in n is negligible in this range [27,28]. However, TLUF is inaccurate for very long wavelengths because the above expression provides *ε*_i_(*E*→0) = ∞. This particular problem has been resolved in the TLU of Rodriguez de Marcos (TLUR) [29,30], which utilizes the expression *ε*_i_(*E*) = const × *E* × exp(*E*/*E*_U_) in the range *E* ≤ *E*_b_, as it provides *ε*_i_(*E*→0) = 0.

Notably, both TLUF and TLUR cannot employ more than one oscillator, which should decrease their accuracy for materials with intricate electronic band structures. The more accurate characterization of such materials could be achieved using DMs with multiple oscillators. In this respect, multiple-oscillator TL, CC, and GA can be included in the universal DM of Franta et al. (UD), which, in principle, allows for film characterization over the entire electromagnetic spectrum [31,32]. In UD, *ε*_r_(*E*) and *ε*_i_(*E*) are obtained by adding, respectively, parametrized electric susceptibility contributions from the individual oscillators, whereas each contribution to *ε*_r_(*E*) is formulated using Kramers–Kronig integration utilizing the corresponding contribution to *ε*_i_(*E*). Moreover, UD can readily be applied to amorphous materials because it can include parametrized electric susceptibility contributions to *ε*_r_(*E*) and *ε*_i_(*E*) from the Urbach tail.

Spectroscopic ellipsometry (SE) and spectrophotometry are primary and complementary techniques for the optical characterization of thin films. SE uses oblique incidence of linearly polarized light, a change in its polarization state due to its reflection from the film, and a selected DM [5,33]. For example, ∆d can be computed via an SE analysis of the depolarization of the reflected light [34,35]. When the film is on a light-transmitting substrate, reflections from the back surface of the substrate are problematic for SE because they create an incoherent additive (without phase information) to the coherent interaction (with phase information) between the light reflected from the two surfaces of the film. This problem can be circumvented by appropriate backside roughening [36] or attaching light scattering refractive index matching material to the back surface of the substrate [37]. Furthermore, the main SE parameters psi-delta have been computed in recent papers [38,39] for a mostly transparent film on a transparent substrate, depending on the spectral bandwidth ∆*λ*, the thickness non-uniformity ∆d of the film, its surface roughness, and its absorption. Nevertheless, it seems that there are no publications reporting the computation of all film characteristics printed in red in Figure 2, in the case of a weakly absorbing substrate, from experimental SE psi-delta spectra and taking into account ∆*λ* (or its respective spectral slit width). The most likely reason for this is the complexity of the phase change in the polarization state of light, during the reflection, depending on all of these film characteristics.

In the spectrophotometry of thin films are measured transmittance spectra *T*(*λ*) and/or reflectance spectra *R*(*λ*), which are most often of a sample consisting of a film deposited on a substrate [40,41]. The spectra *T*(*λ*) and *R*(*λ*) are scanned using a spectrophotometer, which can commonly operate in the UV/Vis/NIR spectral region [14,15]. *T*(*λ*) of such a sample is measured usually at the normal incidence of the light beam to the film and *R*(*λ*) at quasi-normal incidence to the film, both of which result in diminishing sideway leaking of light, disregarding the polarization state of light, and reducing mathematical complications [42,43]. Also, it is preferable to perform film characterization only from *T*(*λ*) or *R*(*λ*) because repositioning the sample for using both of them can lead to not identical light spots on the surface of the film and, therefore, to different averages of film thicknesses over these light spots [40,44].

In this present paper, the optical characteristics of the substrate are designated by the subscript “s”. Moreover, we studied only the quasi-normal incidence reflectance *R*(*λ*) of a thin film on a substrate and the normal incidence transmittance *T*(*λ*) of such a sample. Furthermore, only the common case of a dominantly coherent interaction between the light reflected from the two surfaces of this film is considered, which is formulated as $\bar{d}$ << *λ*^2^/(2*n*∆*λ*), as well as incoherent interaction between the light reflected from the two surfaces of the substrate, i.e., d_s_ >> *λ*^2^/(2*n*_s_∆*λ*) [45]. The reflectance *R*(*λ*) of a thin film with non-uniform thickness, named non-uniform film, on such a substrate, known as a thick planar substrate, is illustrated in Figure 2.

There are several spectrophotometric methods for thin film characterizations employing *R*(*λ*), as shown in Figure 2. For example, *n*(*λ*) and *k*(*λ*) have been calculated from the measurements of *d*, *R*(*λ*), and *T*(*λ*) or the absorbance *A*(*λ*) of the sample, assuming that the film has uniform thickness *d* (such film is named uniform film) and the substrate is infinite [46,47,48,49]. Furthermore, it is well established that *λ*_g_ < 2000 nm (*E*_g_ > 0.6 eV) is valid for the vast majority of semiconductors and dielectrics [50,51], which means that they have a significant spectral range of wavelengths *λ* > *λ*_g_ < 2000 nm with weak absorption and *n*^2^(*λ*) >> *k*^2^(*λ*) << 1 in the UV/Vis/NIR region. The facts from the last two paragraphs indicate that there are apparent minima and maxima in this spectral range of *T*(*λ*) and *R*(*λ*) of the sample from Figure 2 due to thin-film interference, provided that the semiconductor or dielectric film is not too thin, i.e., $\bar{d}$ > *λ*/(2*n*) [45]. In such a case, it is possible to draw two envelopes around the apparent maxima and around the apparent minima of *T*(*λ*) or *R*(*λ*), as well as to apply the envelope method (EM) for film characterization in this spectral range [52,53]. The tangency points between the spectrum *T*(*λ*) or *R*(*λ*) and its two envelopes can be represented by the respective tangency wavelengths *λ*_t_(*l*), where the integer *l* is the tangency wavelength number counted from the longer wavelengths end of the spectrum. Spectra with apparent minima and maxima due to thin-film interference are referred to as interference spectra.

EM is a spectrophotometric method that utilizes an interference fringes equation, including all *λ*_t_(*l*), together with the interference spectrum *T*(*λ*) or *R*(*λ*) and its two envelopes [52,53]. This additional information makes it possible to achieve a higher accuracy of computation of the $\bar{d}$, ∆d, and *n*(*λ*) of the film, in the range of its weak absorption, by using EM in comparison with exclusively DM-based parametrization or SE. In this study, EM for a spectrum *T*(*λ*) is abbreviated as EMT, and EM for *R*(*λ*) is denoted as EMR, whereas the ultimately computed characteristics of the film are designated by the subscript “c”.

The most popular EMT is the method of Swanepool for uniform film [54] and for non-uniform film [55], both developed assuming a finite transparent thick planar substrate with *n*(*λ*) > *n*_s_(*λ*) > 1. Our group enhanced this EMT to account for substrate absorption, as ∆*d*_c_ and the interval *l* ⊂ [*l*_1c_, *l*_2c_] of tangency wavelength numbers participating in the characterization were deduced based on observation [56]. Furthermore, we improved EMT by developing an optimizing envelope method for *T*(*λ*), abbreviated as OEMT [57,58,59]. OEMT employs external smoothing of the noisy interference spectrum *T*(*λ*), which generates externally smoothed interference spectrum *T*_sm0_(*λ*) touching externally *T*(*λ*), instead of common smoothing generating *T*_sm0_(*λ*) passing closer to internally to *T*(*λ*). Such externally smoothed *T*_sm0_(*λ*) is intended to offset the partial coherence of light in the film due to light scattering mainly from the roughness of the surface film/air, and *T*_sm0_(*λ*) is slit width corrected thus providing a spectrum *T*_sm_(*λ*) used thereafter in the film characterization. In essence, OEMT computes optimized values $\bar{d}$_c_ of the average thickness and ∆*d*_c_ of the thickness non-uniformity of the film over the light spot via the minimization of a selected error metric ERM for wavelengths *λ* within an optimized interval [*λ*_t_(*l*_2c_), *λ*_t_(*l*_1c_)] corresponding to an optimized interval [*l*_1c_, *l*_2c_].

The computed results derived from our previous EMT and OEMT characterizations of the thickness parameters of thin films of amorphous semiconductors are displayed in Table 1. The same table also includes the relative error *RE*($\bar{d}$_c_) of the computed average film thickness $\bar{d}$_c_ over the light spot, which corresponds to ERM representing the root mean square error of the estimations of the film thickness calculated for every *l* from the interval [*l*_1c_, *l*_2c_].

It is seen from the data about EMT characterizations and *RE*($\bar{d}$_c_) in Table 1 that the accuracy of the computed average film thickness $\bar{d}$_c_ of the uniform film tends to decrease when the substrate is absorbing in comparison with the transparent substrate. Also, for EMT characterizations, the accuracy of $\bar{d}$_c_ apparently decreases even more for the non-uniform film. The data about *RE*($\bar{d}$_c_) from Table 1 also reveal that using OEMT characterization, with fixed *l*_1_ = 1, even of a non-uniform film on absorbing substrate leads to the accuracy of $\bar{d}$_c_ commensurate with that of EMT characterization of a uniform film on a transparent substrate.

Furthermore, the data from the fourth and fifth text lines in Table 1 indicate that OEMT provides $\bar{d}$_c_ with significantly smaller *RE*($\bar{d}$_c_) compared to its respective EMT. It is also seen that using OEMT with external smoothing and optimized ∆*d*_c_ and [*l*_1c_, *l*_2c_] results in the relative error *RE*($\bar{d}$_c_) < 0.1% in the computation of the average film thickness. This very high accuracy of $\bar{d}$_c_ is due to the very high accuracy of the optimized Δ*d*_c_ and [*l*_1c_, *l*_2c_] because $\bar{d}$_c_, ∆*d*_c_, and [*l*_1c_, *l*_2c_] are obtained at the same step of the OEMT algorithm. In addition to this, it is sensible that employing very accurate $\bar{d}$_c_ and ∆*d*_c_ can lead to the computation of very accurate spectral dependencies *n*_c_(*λ*) and *k*_c_(*λ*) of the film. With respect to the above, it is not surprising that our comparative study [62] showed that OEMT ensures the most accurate characterization of a-Si films, with significantly different average thicknesses, amongst four methods selected as most likely to provide an accurate characterization of such films. Nevertheless, neither external smoothing of interference spectra nor the use of an optimized interval [*l*_1c_, *l*_2c_] of tangency wavelength numbers are available in commercial software for spectroscopic ellipsometry [63,64] or spectrophotometry [65,66].

EMR for thin film characterization only from *R*(*λ*) was proposed in [67] for uniform film on a finite transparent thick planar substrate with *n*(*λ*) > *n*_s_(*λ*) > 1, whereas this EMR, also known as the method of Minkov, has been utilized by research groups in several countries [68,69,70,71,72]. Moreover, some modifications of EMR from [67] have been dedicated to the characterization of semiconductor or dielectric thin film from *R*(*λ*), assuming that the film has uniform thickness and the thick planar substrate is transparent [73,74,75,76,77]. The non-uniformity Δd of the film has been included in formulae for such *R*(*λ*) in [78,79,80], however in none of these papers have been taken into account absorption in the substrate, external smoothing of *R*(*λ*), or optimization of $\bar{d}$_c_, ∆*d*_c_, or [*l*_1c_, *l*_2c_]. As already described, though, accounting for these factors in OEMT results in increasing the accuracy of characterization of a thin film on a thick planar substrate with *n*(*λ*) > *n*_s_(*λ*) > 1. Considering the above, the main goal of this study is to develop and use an optimizing envelope method for the characterization of a thin semiconductor or dielectric film on a finite and generally non-transparent thick planar substrate, only from quasi-normal incidence *R*(*λ*) of the sample. This method, abbreviated as OEMR, has to resolve the above-mentioned deficiencies of the approaches from [67,68,69,70,71,72,73,74,75,76,77,78,79,80]. Another goal is to determine the most accurate method for the characterization of a-Si films with dissimilar thicknesses, only from UV/Vis/NIR spectrum *R*(*λ*), by comparing the accuracies of such characterizations employing different DM-based methods.

## 2. Materials and Methods

### 2.1. Theory

A formula about the reflectance *R*_u_(*λ*) of a uniform thin film on a finite and generally non-transparent (with *k*_s_ > 0 for some *λ*) thick planar substrate was derived in [81] via flow graph theory, assuming coherent interaction between light reflected from the two surfaces of the film and incoherent interaction between light reflected from the two surfaces of the substrate. In the case of quasi-normal incidence of light to the film, this formula is rewritten as follows:(3)$$R_{u}(\lambda )={\rho_{a,s}}^{2}+\frac{\left({n_{s}}^{2}+{k_{s}}^{2}\right){\left({\tau_{a,f}}^{2}{\tau_{f,s}}^{2}\rho_{s,a}x_{s}\right)}^{2}}{1-{\left(\rho_{s,a}^{\prime }\rho_{s,a}x_{s}\right)}^{2}},$$
whereas ${\dot{t}}_{a,f}=\tau_{a,f}\exp(i\xi_{a,f})=2{\dot{N}}_{a}/\left({\dot{N}}_{a}+{\dot{N}}_{f}\right),\;{\dot{t}}_{f,s}=\tau_{f,s}\exp(i\xi_{f,s})=2{\dot{N}}_{f}/\left({\dot{N}}_{f}+{\dot{N}}_{s}\right),$${\dot{r}}_{a,f}=\rho_{a,f}\exp(i\psi_{a,f})=\left({\dot{N}}_{a}-{\dot{N}}_{f}\right)/\left({\dot{N}}_{a}+{\dot{N}}_{f}\right),\;{\dot{r}}_{f,s}=\rho_{f,s}\exp(i\psi_{f,s})=\left({\dot{N}}_{f}-{\dot{N}}_{s}\right)/\left({\dot{N}}_{f}+{\dot{N}}_{s}\right),$${\dot{r}}_{s,a}=\rho_{s,a}\exp(i\psi_{s,a})=\left({\dot{N}}_{s}-{\dot{N}}_{a}\right)/\left({\dot{N}}_{s}+{\dot{N}}_{a}\right),$${\dot{r}}_{a,s}=\rho_{a,s}\exp(i\psi_{a,s})=\frac{{\dot{r}}_{a,f}+{\dot{r}}_{f,s}{\dot{\varsigma }}^{2}}{1+{\dot{r}}_{a,f}{\dot{r}}_{f,s}{\dot{\varsigma }}^{2}},\;{\dot{r}}_{s,a}^{\prime }=\rho_{s,a}^{\prime }\exp(i\psi_{s,a}^{\prime })=\frac{-{\dot{r}}_{f,s}-{\dot{r}}_{a,f}{\dot{\varsigma }}^{2}}{1+{\dot{r}}_{a,f}{\dot{r}}_{f,s}{\dot{\varsigma }}^{2}},\;x_{S}=\exp(-4\pi k_{S}d_{S}/\lambda ),$$\varphi =4\pi nd/\lambda ,\;\alpha =4\pi k/\lambda ,\;x=\exp(-4\pi kd/\lambda )=\exp(-\alpha d),\;\bar{x}=\exp(-\alpha \bar{d}),$$\dot{\varsigma }=\exp(i2\pi \dot{N}d/\lambda )=\exp(-2\pi kd/\lambda )\exp(i2\pi nd/\lambda )=\sqrt{x}\exp(i\varphi /2),$
the subscripts “a” and “f” refer to air and film, respectively, and the superscript ’ indicates backpropagation of light. The expression of *R*_u_(*λ*) in Equation (3) using complex numbers is convenient for development of computer code for optical characterization of thin films. The incoherent interaction between light reflected from the two surfaces of the substrate can be confirmed via absence of interference extrema in reflectance spectrum *R*_s_(*λ*) of the bare substrate (without film on it).


Furthermore, the reflectance spectrum *R*(*λ*) of non-uniform thin film, with uniformly distributed thickness *d* ⊂ [$\bar{d}$ − ∆d,$\bar{d}$ + ∆d] over the light spot, on a finite and generally non-transparent thick planar substrate can be obtained by numerical integration of *R*_u_ [78,82]:(4)$$R\left(\lambda \right)=\frac{1}{\varphi_{2}-\varphi_{1}}\int_{\varphi_{1}}^{\varphi_{2}}R_{u}d\varphi ,$$
where $\varphi_{1}=4\pi n(\bar{d}-\Delta d)/\lambda ,\varphi_{2}=4\pi n(\bar{d}+\Delta d)/\lambda$. Equation (4) takes into account both the substrate absorption and the finite size of the thick substrate, which represents a novelty in spectrophotometry of thin non-uniform films.

On the other hand, envelopes can be computed for every interference reflectance spectrum with at least three apparent maxima and three apparent minima, whereas the envelope along the maxima is designated as *R*_+_(*λ*), and the envelope along the minima is *R*_−_(*λ*) [83]. Also, the tangency wavelengths *λ*_t_ between the smoothed and slit width corrected reflectance spectrum *R*_sm_(*λ*) and its envelopes *R*_+_(*λ*) and *R*_−_(*λ*) should satisfy the interference fringes equation. In the region of weak and medium absorption in the film, and *n*(*λ*) > *n*_s_(*λ*) > 1, which is commonly valid for a semiconductor or dielectric film, e.g., on a glass substrate [84,85], the interference fringe equation is expressed as follows [78,82]:(5)$$\begin{array}{l} 2n(\lambda_{t})\bar{d}=m_{l}(\lambda_{t})\lambda_{t}(l)\\ \mathrm{where}\left\{ \begin{array}{l} m_{l}\geq 1/2—\mathrm{half}-\mathrm{integer}\;\mathrm{for}\;\mathrm{all}\;\mathrm{tangency}\;\mathrm{wavelengths}\;\lambda_{t}(l)\;\mathrm{from}\;\mathrm{the}\;\mathrm{envelope}R_{+}(\lambda ),\\ m_{l}\geq 1—\mathrm{integer}\;\mathrm{for}\;\mathrm{all}\;\mathrm{tangency}\;\mathrm{wavelengths}\;\lambda_{t}(l)\;\mathrm{from}\;\mathrm{the}\;\mathrm{envelope}R_{-}(\lambda ), \end{array}\right. \end{array}$$
*l* = 1, 2, … *l*_M_ is the tangency wavelength number counted from the longer wavelengths end of *R*(*λ*), and $m_{l}(\lambda_{t})$ is the interference order.

Moreover, the following expression calculates the estimated film thickness:(6)$$d_{1}\left(l\right)=\bar{d}\left[\lambda_{t}\left(l\right)\right]=\frac{\lambda_{t}\left(l\right)\lambda_{t}\left(l+1\right)}{4\left[\lambda_{t}\left(l\right)n\left(l+1\right)-\lambda_{t}\left(l+1\right)n\left(l\right)\right]}$$
Equation (6) is obtained by rewriting Equation (5) for a pair of adjacent tangency wavelengths *λ*_t_(*l*) and *λ*_t_(*l* + 1) corresponding to a pair of tangency points *R*_sm_[*λ*_t_(*l*)] and *R*_sm_[*λ*_t_(*l* + 1)] between *R*_sm_(*λ*) and its envelopes (one of these points is from the envelope *R*_+_(*λ*), and the other is from the envelope *R*_−_(*λ*)) [45,82]. Therefore, the average film thickness $\bar{d}$ over the light spot can be approximated using Equations (5) and (6) in the region of weak and medium absorption in the film.

Taking into account Equation (5), the variable φ from Equation (3) can be expressed as follows for the envelope *R*_+_(*λ*):(7)$$\varphi =4\pi n\left[\bar{d}+\left(d-\bar{d}\right)\right]/\lambda \overset{\mathrm{for}R_{+}(\lambda )}{\underset{\mathrm{from}\;\mathrm{Equation}\;(5)}{\to }}=2\pi .(\mathrm{half}-\mathrm{integer})+4\pi n\left(d-\bar{d}\right)/\lambda .$$

Since φ participates in Equations (3) and (4) only via the functions sin(φ) and cos(φ) (as seen in the lines of Equation (3), including φ and $\dot{\varsigma }$), which are periodical functions with a period of 2π, a formula for the upper envelope *R*_+_(*λ*) of *R*_sm_(*λ*) is derived by substituting Equation (7) in Equation (4) as follows:(8)$$R_{+}(\lambda )=\frac{1}{\varphi_{2_{+}}-\varphi_{1_{+}}}\int_{\varphi_{1_{+}}}^{\varphi_{2_{+}}}R_{u}(\varphi_{+})d\varphi_{+}=\frac{1}{\Delta \varphi }\int_{\varphi_{1_{+}}}^{\varphi_{2_{+}}}R_{u}(\varphi_{+})d\varphi_{+},$$
where
$$\varphi_{+}=4\pi n(d-\bar{d})/\lambda +\pi ,\;\varphi_{1_{+}}=-4\pi n\Delta d/\lambda +\pi ,\;\varphi_{2_{+}}=4\pi n\Delta d/\lambda +\pi ,\;\Delta \varphi =\varphi_{2_{+}}-\varphi_{1_{+}}=8\pi n\Delta d/\lambda .$$

Similarly, φ can be expressed, also from Equation (5), for the envelope *R*_(*λ*) as follows:(9)$$\varphi =4\pi n\left[\bar{d}+\left(d-\bar{d}\right)\right]/\lambda \overset{\mathrm{for}\;R\_(\lambda )}{\underset{\mathrm{from}\;\mathrm{Equation}\;(5)}{\to }}=2\pi .(\mathrm{integer})+4\pi n\left(d-\bar{d}\right)/\lambda .$$

Correspondingly, a substitution of Equation (9) in Equation (4) provides a formula for the lower envelope *R*_(*λ*) of *R*_sm_(*λ*) as follows:(10)$$R_{-}(\lambda )=\frac{1}{\varphi_{2_{-}}-{\varphi_{1}}_{-}}\int_{\varphi_{1_{-}}}^{\varphi_{2_{-}}}R_{u}(\varphi_{-})d\varphi_{-}=\frac{1}{\Delta \varphi }\int_{\varphi_{1_{-}}}^{\varphi_{2_{-}}}R_{u}(\varphi_{-})d\varphi_{-},$$
where



$\varphi_{-}=4\pi n(d-\bar{d})/\lambda ,\;{\varphi_{1}}_{-}=-4\pi n\Delta d/\lambda ,\;{\varphi_{2}}_{-}=4\pi n\Delta d/\lambda ,\;\Delta \varphi ={\varphi_{2}}_{-}-{\varphi_{1}}_{-}=8\pi n\Delta d/\lambda .$



Furthermore, the following analytic approximations of the envelopes *R*_+_(*λ*) and *R*_−_(*λ*) are obtained by substituting *R*_u_(*λ*) from Equation (3) into Equations (8) and (10) and solving the respective symbolic integrals as follows:(11)$$\begin{array}{l} R_{0^{\pm }}(\lambda )≃1-\left[\frac{a_{3_{0}}-a_{2_{0}}}{\theta \sqrt{{a_{3_{0}}}^{2}-{b_{3_{0}}}^{2}}}-\frac{\left(n_{s}^{2}+k_{s}^{2}\right){\left({\tau_{a,f_{0}}}^{2}{\tau_{f,s_{0}}}^{2}\rho_{s,a}\bar{x}x_{s}\right)}^{2}b_{3_{0}}}{\theta (a_{1_{0}}b_{3_{0}}-a_{3_{0}}b_{1_{0}})\sqrt{{a_{3_{0}}}^{2}-{b_{3_{0}}}^{2}}}\right]\tan^{-1}\left[\sqrt{\frac{a_{3_{0}}\mp b_{3_{0}}}{a_{3_{0}}\pm b_{3_{0}}}}\tan\left(\theta \right)\right]+\\ -\frac{\left(n_{s}^{2}+k_{s}^{2}\right){\left({\tau_{a,f_{0}}}^{2}{\tau_{f,s_{0}}}^{2}\rho_{s,a}\bar{x}x_{s}\right)}^{2}b_{1_{0}}}{\theta \left(a_{1_{0}}b_{3_{0}}-a_{3_{0}}b_{1_{0}}\right)\sqrt{{a_{1_{0}}}^{2}-{b_{1_{0}}}^{2}}}\tan^{-1}\left[\sqrt{\frac{a_{1_{0}}\mp b_{1_{0}}}{a_{1_{0}}\pm b_{1_{0}}}}\tan\left(\theta \right)\right], \end{array}$$
where
$$\begin{array}{c} a_{1_{0}}=1-{\left(\rho_{a,f_{0}}\rho_{s,a}\bar{x}x_{S}\right)}^{2}+{\left(\rho_{a,f_{0}}\rho_{f,s_{0}}\bar{x}\right)}^{2}-{\left(\rho_{f,s_{0}}\rho_{s,a}x_{s}\right)}^{2},\;b_{1_{0}}=2\rho_{a,f_{0}}\rho_{f,s_{0}}\bar{x}\left[1-{\left(\rho_{s,a}x_{S}\right)}^{2}\right],\\ a_{2_{0}}={\rho_{a,f_{0}}}^{2}+{\left(\rho_{f,s_{0}}\bar{x}\right)}^{2},\;b_{3_{0}}=2\rho_{a,f_{0}}\rho_{f,s_{0}}\bar{x},\;a_{3_{0}}=1+{\left(\rho_{a,f_{0}}\rho_{f,s_{0}}\bar{x}\right)}^{2},\;\theta =2\pi n\Delta d/\lambda =\Delta \varphi /4,\\ \tau_{a,f_{0}}=\frac{2}{\left(n+1\right)},\;\tau_{f,s_{0}}=\frac{2n}{\sqrt{{\left(n+n_{s}\right)}^{2}+{k_{s}}^{2}}},\;\rho_{a,f_{0}}=\frac{n-1}{n+1},\;\rho_{f,s_{0}}=\sqrt{\frac{{\left(n-n_{s}\right)}^{2}+{k_{s}}^{2}}{{\left(n+n_{s}\right)}^{2}+{k_{s}}^{2}}}, \end{array}$$
as the upper sign from the “±” and “∓” symbols corresponds to *R*_+_(*λ*), and the lower of these signs refer to *R*_−_(*λ*). Equation (11) is derived by replacing the transmittance x=exp(−αd) in the film (for light passing once between its two surfaces) with its averaging transmittance $\bar{x}=\exp(-\alpha \bar{d})$ over the light spot, and ignoring *k*(*λ*) in the Fresnel coefficients denoted by “*τ*” and “*ρ*” in Equation (3). The novel Equation (11) makes it possible to expand the EMR framework to account for both the substrate absorption and the finite size of the thick planar substrate, unlike the respective expressions used in EMR from [67,68,69,70,71,72,73,74,75,76,77,78].

### 2.2. Features of Simulated Quasi-Normal Incidence Interference Spectra R(λ) and Their Envelopes

Several model specimens corresponding to a-Si:H film on a thick planar glass substrate are introduced for representing the typical behavior of interference reflectance spectra *R*(*λ*) of a thin semiconductor film on a thick planar substrate. The refractive index and the extinction coefficient of the model film are $n\left(\lambda \right)=2.6+3\times 10^{5}/\lambda^{2}$ and $k\left(\lambda \right)=\left(\lambda /4\pi \right)\times 10^{1.5\times 10^{6}/\lambda^{2}-8}$, where *λ* (nm) [54]. The refractive index *n*_s_(*λ*) of the model substrate is that of a standard microscope slide glass substrate G50 of Levenhuk [83,84], and its extinction coefficient is either *k*_s_(*λ*) > 0 of the same substrate of Levenhuk or *k*_s_(*λ*) = 0. Interference reflectance spectra of the model specimens are calculated from Equation (3) for uniform film and from Equation (4) for non-uniform film, and interference transmittance spectrum *T*_u_(*λ*) of one model specimen with uniform film is calculated as in [59,86]. Such simulated reflectance spectra and transmittance spectrum, their envelopes and corresponding absorbance spectrum of the model specimens, and other optical characteristics of their films and substrates, are presented in Figure 3a,b,d. Figure 3c shows differences ∆*R* between reflectance spectra as well as differences ∆*R*_+_ and ∆*R*_−_ between their respective envelopes, computed either by numerical integration (NI) with N_st_ integration steps from Equations (4), (8) and (10) or by the analytic approximations from Equation (11).

Analysis of the graphs from Figure 3 reveals the following dependencies:(2.2.1)*R*_u_(*λ*) + *T*_u_(*λ*) ≈ 1 in the region of weak absorption in the film because the absorbance of the specimen with transparent substrate is *A*_u_(*λ*) = 1 − *R*_u_(*λ*) + *T*_u_(*λ*) ≈ 0 in this region, which is illustrated with the dashed magenta line in Figure 3a. Therefore, the interference patterns of *R*(*λ*) and *T*(*λ*) have very similar features in the region of weak absorption in the film. This fact indicates that advances used for accurate characterization of a thin film from *T*(*λ*) (such as those of OEMT), in the region of weak absorption in the film, can be readily employed for accurate characterization of a thin film from *R*(*λ*). Moreover, it is seen also from Figure 3a that the reflectance spectrum of the specimen shrinks, and its lower envelope *R*_u−_(*λ*) drifts above the reflectance *R*_s_(*λ*) of the bare substrate with decreasing *λ* in the region of medium absorption in the film, where *A*_u_(*λ*) rises.(2.2.2)The lower envelope *R*_u−_(*λ*) of *R*(*λ*) almost coincides with *R*_s_(*λ*) in the region of weak absorption in uniform film (Figure 3a), and *R*_−_(*λ*) is positioned above *R*_s_(*λ*) in the region of weak absorption in non-uniform film (Figure 3b). However, such a drift of *R*_−_(*λ*) above *R*_s_(*λ*) can be due to both thickness non-uniformity of the film and absorption in the film, taking into account the last comment from (2.2.1). On the other hand, it was indicated in the introduction that OEMT renders optimized values of both the average thickness $\bar{d}$ and the thickness non-uniformity ∆d of the film based on analysis for its region of weak absorption. As a result of the above and the principle similarity with OEMT, it is expected that employing OEMR can lead to superior accuracy characterization of semiconductor and dielectric films, only from *R*(*λ*), over the entire UV/Vis/NIR *R*(*λ*).(2.2.3)In the region where the averaging transmittance in the film is $\bar{x}=\exp(-\alpha \bar{d})$ ≈ 0, the film is opaque, the envelopes *R*_+_(*λ*) and *R*_−_(*λ*) merge (as seen in Figure 3b), the incident light is reflected only at the surface film/air, and *R*(*λ*) = *ρ*_a,f_^2^(*λ*) = [(*n* − 1)^2^ + *k*^2^]/[(*n* + 1)^2^ + *k*^2^]. Therefore, the interference-free part of *R*(*λ*) supplies information about *n*(*λ*) and *k*(*λ*), which can be used for their accurate determination in this region, unlike the transmittance spectrum because *T*(*λ*) ≈ 0 there.(2.2.4)Regarding Figure 3c, the absolute values of all differences in reflectance, calculated by NI with N_st_ = 100 and N_st_ = 30, do not exceed 10^−4^. Since 10^−4^ is quite a small value compared to *R*(*λ*) and the maximum difference in thicknesses *d* over the light spot of this model film is 2∆d = 60 nm, all NI in Equations (4), (8) and (10) are executed using N_st_ ≥ ∆d(nm) ∗ 30/60 = ∆d(nm)/2 from here on in this paper. Furthermore, the absolute values of the differences ∆*R*_+_ and ∆*R*_−_, calculated by NI with N_st_ = 100 and the analytical approximation from Equation (11), do not exceed 2 × 10^−5^ over the region of weak absorption in the film, which exemplifies the accuracy of Equation (11).(2.2.5)The lower envelope *R*_−_(*λ*) of *R*(*λ*) is more dependent on the substrate characteristics than the upper envelope *R*_+_(*λ*), as comprehended from Figure 3d. Correspondingly, the envelope *R*_+_(*λ*) should be more dependent on the film characteristics than the envelope *R*_−_(*λ*), which indicates that employing *R*_+_(*λ*) is more likely to provide accurate film characteristics compared to *R*_−_(*λ*). The appearances of *R*(*λ*) and its envelopes in Figure 3d also clarify that presence of only two apparent extrema of *R*(*λ*) at each of its envelopes should not be sufficient for precise calculation of *R*_+_(*λ*) and *R*_−_(*λ*).(2.2.6)There are notable differences in *R*(*λ*) and its two envelopes for *k*_s_ > 0 and *k*_s_ = 0, respectively, in the region of significant absorption in the substrate, enclosed by a red colored rectangle in Figure 3b,d. Therefore, taking into account the absorption in the substrate should result in increasing the accuracy of characterization of a thin film only from *R*(*λ*).

### 2.3. The Algorithm of OEMR and Its Details

The OEMR is developed for the common case of quasi-normal incidence of light to a specimen consisting of a thin semiconductor or dielectric film on a thick planar substrate with *n*(*λ*) > *n*_s_(*λ*) > 1. The algorithm of OEMR is based on the algorithms of EMR from [78] and OEMT from [57,58], and its steps are shown in Figure 4.

In this present study, at step A2 of the algorithm can be independently employed three different kinds of initial smoothing of *R*(*λ*) (needed for removing its false extrema due to noise), providing a smoothed spectrum *R*_sm0_(*λ*) in the region of the interference pattern. More specifically, “internally smoothed” *R*_sm0_(*λ*) touches internally the noisy spectrum *R*(*λ*), “externally smoothed” *R*_sm0_(*λ*) touches externally *R*(*λ*), and “medium smoothed” *R*_sm0_(*λ*) passes in the middle of the previous two.

The smoothed spectrum *R*_sm_(*λ*) is obtained, at step A3, using a slit width correction of *R*_sm0_(*λ*), based on the fact that a similar correction has been employed for *T*(*λ*) in [54,57] as follows:(12)$$R_{\mathrm{sm}}(\lambda_{t})=R_{\mathrm{sm}0}(\lambda_{t})\pm {\left[\frac{R_{\mathrm{sm}0}(\lambda_{t})\;\Delta s}{\Delta l(\lambda_{t})}\right]}^{2}$$
where Δs is the spectral slit width, and $\Delta l\left(\lambda_{t}\right)=\Delta l\left[\lambda_{t}(l)\right]=\lambda_{t}(l-1)\;-\;\lambda_{t}(l+1)$ is the line width at *λ*_t_(*l*), as “+” from the “±” symbol refers to a maximum of *R*_sm0_(*λ*) and “−” to its minimum. In general, slit width correction is needed for thicker film samples with some linewidths Δ*l*(*λ*_t_) < 10Δs in their *R*(*λ*) [87]. At step A4 of the algorithm, the envelopes *R*_+_(*λ*) and *R*_−_(*λ*) of *R*_sm_(*λ*) are computed using left and right “boundary points”, as described in [88,89] as well as the interpolation procedure from [58], for each of the two envelopes. The envelopes around internally smoothed *R*_sm_(*λ*) are named “internal envelopes of *R*(*λ*)”, the envelopes around externally smoothed *R*_sm_(*λ*)—“external envelopes of *R*(*λ*)”, and the envelopes around medium smoothed *R*_sm_(*λ*)—“medium envelopes of *R*(*λ*)”.

Part B of the algorithm of OEMR, presented in Figure 5, performed optimization, providing optimized results for the interval [*l*_1c_, *l*_2c_] of tangency wavelength numbers, the lowest interference order *m*_1c_ = *m_l_*[*λ*_t_(*l* = 1)] = *m_l_*(*l* = 1), the thickness non-uniformity ∆*d*_c_ of the film, and its average thickness $\bar{d}$_c_ over the light spot.

With respect to steps B1 and B12 of the algorithm, the performance of seven error metrics (*ERM*) for employment in OEMT from [58,59] (only for *T*(*λ*)) have been investigated in [90]. It has been concluded there that the presented below error metrics *ERM*_1_ and *ERM*_2_ provide the most accurate OEMT characterization for a wide variety of semiconductor and dielectric thin films on a finite generally non-transparent thick planar substrate:(13)$$\begin{array}{l} ERM_{1}\left(i_{1},l_{1},l_{2}\right)=\frac{1}{l_{2}-l_{1}}\sqrt{\frac{\sum_{l=l_{1}}^{l_{2}-1}{\left[m_{l}\left(i_{1},l\right)-m_{e}\left(i_{1},l\right)\right]}^{2}}{l_{2}-l_{1}}}\\ =\frac{1}{N_{l}\left(l_{1},l_{2}\right)-1}\sqrt{\frac{\sum_{l=l_{1}}^{l_{2}-1}{\left[m_{l}\left(i_{1},l\right)-m_{e}\left(i_{1},l\right)\right]}^{2}}{l_{2}-l_{1}}}=\frac{RMSE\left(m_{e}\right)}{N_{l}-1}\geq 0,\\ ERM_{2}\left(i_{1},l_{1},l_{2}\right)=\frac{1}{l_{2}-l_{1}+1}\sqrt{\frac{\sum_{l=l_{1}}^{l_{2}}{\left[{\bar{d}}_{e}\left(i_{1},l_{1},l_{2}\right)-d_{2}\left(i_{1},l\right)\right]}^{2}}{l_{2}-l_{1}+1}}\\ =\frac{1}{N_{l}\left(l_{1},l_{2}\right)}\sqrt{\frac{\sum_{l=l_{1}}^{l_{2}}{\left[{\bar{d}}_{e}\left(i_{1},l_{1},l_{2}\right)-d_{2}\left(i_{1},l\right)\right]}^{2}}{l_{2}-l_{1}+1}}=\frac{\sigma ({\bar{d}}_{e})}{N_{l}}\geq 0,\\ ERM_{i}\left(i_{1},l_{1},l_{2}\right)=ERM_{1}\left(i_{1},l_{1},l_{2}\right)\;\mathrm{or}\;ERM_{2}\left(i_{1},l_{1},l_{2}\right), \end{array}$$
where *RMSE* means root mean square error, *σ* designates standard deviation, and $N_{l}\left(l_{1},l_{2}\right)$ is the number of tangency wavelengths in the interval [*l*_1_, *l*_2_]. The suitability of *ERM*_i_ (from Equation (13)) for OEMR is justified via the validity of the approximation *T*(*λ*) + *R*(*λ*) ≈ 1 in the region of weak absorption in the film, as illustrated in Figure 3a and described in paragraph (2.2.1). A principle advantage of *ERM*_i_ is that it provides a wider interval [*l*_1_, *l*_2_], over which the computed reflectance spectrum *R*_c_(*λ*) (obtained via substitution of *n*_c_(*λ*), *k*_c_(*λ*), $\bar{d}$_c_, and ∆*d*_c_ in Equation (4)) represents a good approximation of the measured *R*(*λ*) compared to, e.g., the error metric *σ*($\bar{d}$_e_) from the numerator of *ERM*_2_.

Moreover, the error metric used at step B15 of the algorithm is expressed as follows:(14)$$\begin{array}{l} ERM_{a}\left(l_{1}\right)=\min\left\{ERM_{i}\left[i_{1}\left(l_{1}\right),\mathrm{fixed}\;l_{1},l_{2}\left(l_{1}\right)\right]\right\}\overset{\mathrm{provides}\;i_{1}\left(l_{1}\right),l_{1},l_{2}\left(l_{1}\right)}{\underset{\mathrm{and}\;\mathrm{their}\;\mathrm{respective}\;}{\to }}\\ m_{1a}\left(l_{1}\right)=m_{e}\left[i_{1}\left(l_{1}\right),l=1\right],\;\Delta d_{a}\left(l_{1}\right)=\Delta d\left[i_{1}\left(l_{1}\right)\right],\;{\bar{d}}_{a}\left(l_{1}\right)={\bar{d}}_{e}\left[i_{1}\left(l_{1}\right),\mathrm{fixed}\;l_{1},l_{2}\left(l_{1}\right)\right]. \end{array}$$

Furthermore, the relative error in the computed average film thickness $\bar{d}$_c_ is expressed by *ERM*_c_ for *ERM*_2_, obtained at step B17, as follows:(15)$$\begin{array}{l} RE({\bar{d}}_{c})=\frac{\sigma \left({\bar{d}}_{c}\right)}{{\bar{d}}_{c}}=\frac{ERM_{c}(ERM_{2})\left(l_{2c}-l_{1c}+1\right)}{{\bar{d}}_{c}}\geq 0\overset{\mathrm{provides}\;l_{1c},l_{2c}\left(l_{1c}\right)}{\underset{\mathrm{and}\;\mathrm{its}\;\mathrm{respective}}{\to }}\\ \left[\lambda_{t}\left(l_{1c}\right),\;\lambda_{t}\left(l_{2c}\right)\right]\to \left[l_{1c},\;l_{2c}\right],m_{1c}=m_{1a}\left(l=1\right),\;\Delta d_{c}=\Delta d_{a}\left(l_{1c}\right),\;{\bar{d}}_{c}={\bar{d}}_{a}\left(l_{1c}\right). \end{array}$$

The optimization in part B of the algorithm of OEMR is achieved by a direct comparison between the error metric values corresponding to all credible sets (*l*_1_,*l*_2_,Δ*d*) rather than using a standard minimization algorithm, which can terminate in a local minimum of the error metric. This is carried out by including an independent loop for every one of the parameters *l*_1_, *l*_2_, and Δ*d*, as seen in Figure 5. There is, therefore, only one possible set of thickness parameters ∆*d*_c_ and $\bar{d}$_c_ of the film, corresponding to the global minimum of *ERM*_i_(*l*_1_,*l*_2_,Δ*d*), as these thickness parameters are computed at the end of part B of the algorithm. Notably, the algorithm of OEMR is validated for the model a-Si:H film with *n*(*λ*) and *k*(*λ*) introduced in the first paragraph from Section 2.2, $\bar{d}$ = 1000 nm and Δd = 30 nm on 3.28 mm thick Borofloat33 substrate, as it turns out that the errors in both $\bar{d}$_c_ and ∆*d*_c_ do not exceed 0.025%.

Furthermore, the computed spectral dependencies *n*_c_(*λ*) for the refractive index and *k*_c_(*λ*) for the extinction coefficient of the film can be determined, as described in part C of the algorithm of OEMR from Figure 4, similarly to using OEMT [57,58]. Moreover, *n*_c_(*λ*) and *k*_c_(*λ*) can be also derived via some existing broad spectrum dispersion models, however employing the already accurately computed via OEMR thickness parameters $\bar{d}$_c_ and ∆*d*_c_ of the film. After $\bar{d}$_c_, ∆*d*_c_, *n*_c_(*λ*), *k*_c_(*λ*), and their respective spectrum *R*_c_(*λ*) from Equation (4) are computed, for all wavelengths from the measured *R*(*λ*), the figure of merit is calculated as follows:(16)$$FOM=1000\times \sqrt{\frac{\sum_{j=1}^{N_{j}}{\left\{R\left[\lambda (j)\right]-R_{c}\left[\lambda (j)\right]\right\}}^{2}}{N_{j}}}=1000\times RMSE\left(R_{c}\right)\geq 0$$
Equation (16) can be utilized as a measure of closeness of *R*_c_(*λ*) to *R*(*λ*), where N*_j_* is the number of all *λ* ⊂ [min(*λ*), max(*λ*)]. Therefore, the film characterization providing smallest FOM is considered to be the most accurate characterization of that film.

## 3. Results

### 3.1. Summary of the Preparation of Samples and the Measurement of Their Reflectance Spectra

a-Si thin films with dissimilar thicknesses were deposited on non-transparent glass substrates via RF-magnetron sputtering using applied RF power of 525 W, target-to-substrate distance of 6.1 cm, and Ar gas. Sample SP1 consists of a-Si film sputtered at Ar pressure of 4.4 Pa on 0.9 mm thick Corning7059 substrate, and the sample SP2 represents a-Si film sputtered at Ar pressure of 0.13 Pa on 3.28 mm thick Borofloat33 substrate. The deposition rate was 73.8 nm/min for the film SP1 (from the sample SP1) and 90.0 nm/min for the film SP2 (from the sample SP2). The reflectance spectra *R*(*λ*) of these samples were measured by a Perkin-Elmer Lambda 1050 UV/visible/NIR double-beam spectrophotometer using specular reflectance accessory B0086703 with Φ = 6° [91,92], 2 nm slit width, 1 nm data collection step, and 10 mm × 3 mm illuminated area of the film. The a-Si thin films SP1 and SP2 are characterized next, only from their corresponding *R*(*λ*), using OEMR and several dispersion models.

### 3.2. Characterizations of the a-Si Films by OEMR

The two a-Si films SP1 and SP2 are characterized via OEMR employing internal envelopes, external envelopes, or medium envelopes of their reflectance spectra *R*(*λ*). The results from the execution of parts A and B from the algorithm of OEMR for these films are presented in Table 2. Data from OEMR characterizations using the deconvolution of these *R*(*λ*) and their respective deconvolution envelopes are also included in Table 2. The deconvolutions are performed assuming Gaussian distribution, with spectral slit width Δs, of the spectrum of light incident on the film.

It is seen from the last two columns of Table 2 that the most accurate results for $\bar{d}$_c_ are achieved by employing the external envelopes of *R*(*λ*), in which case, the differences in $\bar{d}$_c_ and ∆*d*_c_ obtained via the error metrics *ERM*_1_ and *ERM*_2_ do not exceed 0.06% and 0.9%, respectively, for both studied films. Therefore, the optimized thickness parameters $\bar{d}$_c_ and ∆*d*_c_ of a given film are derived by using OEMR with the external envelopes of *R*(*λ*), and the values of these parameters are selected to be equal to half of the sums of their corresponding values obtained via *ERM*_1_ and *ERM*_2_.

Reflectance spectra, including illustrations of the smoothing and the boundaries of the intervals [*l*_1c_, *l*_2c_], corresponding to the optimized thickness parameters $\bar{d}$_c_ and ∆*d*_c_ from Table 2, are shown in Figure 6, for both studied samples. The dashed red lines in Figure 6c,d represent the computed reflectance spectra *R*_c_(*λ*) corresponding to the characterizations of the films SP1 and SP2 via the synthetic dispersion model “OEMR with 3NA” described in Section 3.5.

Taking into account the interference fringes in Equation (5), the significantly larger number of interference extrema of *R*(*λ*) in Figure 6d compared to Figure 6c confirms that the film SP2 is much thicker than the film SP1. Furthermore, the larger difference *R*_−_(*λ*) − *R*_s_(*λ*) > 0 from Figure 6d than from Figure 6c reaffirms that the film SP2 has larger thickness non-uniformity ∆d compared to the film SP1, according to the description from paragraph (2.2.2). The fact that *l*_1c_ > 1, illustrated in both Figure 6c,d, indicates that OEMR tends to disregard a small number of the largest wavelengths extrema of *R*_sm_(*λ*). This should be related to the unavoidable small inaccuracy of the right boundary points *R*_+_[max(*λ*)] and *R*_−_[max(*λ*)] from the envelopes *R*_+_(*λ*) and *R*_−_(*λ*).

Images regarding the execution of part C from the OEMR algorithm are presented in Figure 7 for both studied films. Each of the Wemple–DiDomenico plots from Figure 7a,c includes green circles corresponding to all *n*_c_(*λ*_t_) calculated at step C1 from the OEMR algorithm, as well as a representation of their linear regression performed over a selected interval [*l*_1w_, *l*_2w_] and depicted by a solid green line. The computed energy *E*_0_ and strength *E*_d_ of the respective undamped single oscillator, as well as the static refractive index *n*(0), are also printed in these plots, whereas the expansion of the linear regression beyond the interval [*l*_1w_, *l*_2w_] is illustrated with a dashed magenta line. The computed refractive index *n*_c_(*λ*) is derived via such an expansion of the linear regression, for wavelengths *λ* within the interval [*λ*_t_(*l*_2w_), max(*λ*)], as described in step C2 from the OEMR algorithm. Furthermore, the extinction coefficients *k*_c_(*λ*) from Figure 7b,d are computed, for wavelengths *λ* also within the interval [*λ*_t_(*l*_2w_), max(*λ*)], using *n*_c_(*λ*) as well as Equations (8) and (10) about the envelopes *R*_+_(*λ*) and *R*_−_(*λ*). The figures of merit *FOM*, printed in Figure 7b,d, are calculated using Equation (16), however applied to the above interval of *λ*.

It is concluded from Figure 6c,d and Figure 7a,c and Equation (1) that DM of Wemple–DiDomenico, corresponding to the undamped single oscillator, describes well *n*(*λ*) in the interval [*l*_1w_, *l*_2w_], representing the region of weak to medium absorption in the a-Si films. In Figure 7a, the small shifts of the green circles from the dashed magenta line for *l* < *l*_1w_ = 3 should be related to the already mentioned small inaccuracy of the envelopes *R*_+_(*λ*) and *R*_−_(*λ*) around their right boundary points *R*_+_[max(*λ*)] and *R*_−_[max(*λ*)]. Moreover, the drift of the green circles away from the dashed magenta line for *l* > *l*_2w_ is due to the inaccuracy of DM of Wemple–DiDomenico, which does not include *k*(*λ*), in the region of strong to medium absorption in the film. Furthermore, the smallest FOMs correspond to the green curves from Figure 7b,d, which indicates that *k*(*λ*) is computed most accurately from *R*_+_(*λ*) and its respective Equation (8) compared to *R*_−_(*λ*) or [*R*_+_(*λ*) + *R*_−_(*λ*)]/2. Nevertheless, the inaccuracy of representation of *n*(*λ*) via the linear regression for *l* > *l*_2w_ from the Wemple–DiDomenico plot indicates that the accuracy of computation of *k*(*λ*) from *R*_+_(*λ*) should decrease with decreasing *λ* in the region of strong to medium absorption in the film. Based on the above comments from this paragraph, it is expected that the accuracy of film characterization over the entire UV/Vis/NIR *R*(*λ*) can be increased by combining OEMR with DM describing more accurately $\dot{N}\left(\lambda \right)$ of the film in its region of medium and strong absorption.

Some amorphous materials can contain voids [93,94], and the voids volume fraction (with respect to the entire volume of the material) has been approximated as follows:(17)$$f_{\mathrm{void}}\;≃\;\frac{\left[1+2n^{2}(0)\right]\left[n_{\mathrm{dense}}^{2}(0)-n^{2}(0)\right]}{3n^{2}(0)\left[n_{\mathrm{dense}}^{2}(0)-1\right]}$$
where *n*_dense_(0) is the static refractive index of the same amorphous material but without voids [60,93]. On the other hand, *E*_0_ ≈ 4.0 and *E*_d_ ≈ 44.4 for crystalline Si [95], whereby *n*(0) ≈ 3.48 is obtained from Equation (1) applied to this material, while the difference between *n*_dense_(0) for a-Si without voids and *n*(0) for crystalline Si is ≈ 0.28 based on data from [96]. Correspondingly, the static refractive index of a-Si without voids is estimated to be *n*_dense_(0) ≈ 3.48 + 0.28 = 3.76. Furthermore, replacing the values of *n*(0), printed in Figure 7a,c, in Equation (17), provides *f*_void_ ≈ 5.4% for the film SP1 and *f*_void_ ≈ 0.4% for the film SP2. Therefore, the somewhat larger value of *k*_c_(*λ*) in the region of weak absorption in the film SP1 (compared to the film SP2) should be due to light reflections from boundaries of such voids, which create incoherent additive to *R*(*λ*), thus slightly smearing its interference pattern. Moreover, effective thickness filled with voids in a film can be expressed as $\bar{d}$_void_ = $\bar{d}$_c_
*f*_void_, which leads to $\bar{d}$_void_ = 36.4 nm for the film SP1 and $\bar{d}$_void_ = 15.4 nm for the film SP2.

### 3.3. Characterizations of the Films via Single Electron Oscillator DMs

The a-Si films are characterized using broad spectrum single electron oscillator DMs 1TL, 1NA, TLUR, and TLUF as well as Equation (4), which includes implicitly the film thickness parameters $\bar{d}$ and ∆d. The computed results from these characterizations are presented in Table 3, where *ε*_r_(∞) and *n*_c_(∞) are the complex dielectric function and the complex refractive index of the film for photon energy *E*→∞, assuming that *k*_r_(∞) = 0. The first three columns of this Table present the values of the oscillator parameters, as the first of them corresponds to the oscillator amplitude, the second is the central energy, and the third is the broadening term. Each symbol of these parameters contains subscripts, of which the first two letters represent the employed DM, and the number after them is the number of the oscillator, i.e., 1 for the DMs featured in Table 3. The initial approximations of the oscillator parameters are selected to be in the midst of the intervals of recommended values for these parameters prescribed by their corresponding DM. In this paper, all film characterizations employing broad spectrum oscillator DMs are performed via the minimization of FOM from Equation (16), separately utilizing Nelder–Mead simplex direct search algorithm and interior point algorithm [97,98]. The differences are recorded between the values of FOM at every two successive steps of the minimization, and the respective film characterization is considered to be completed when the absolute values of these differences fall below 0.001 for both of these algorithms.

The data from Table 3 indicate that TLUF provides the most accurate characterization of both a-Si films (i.e., characterization with the smallest FOM) amongst the four broad-spectrum single oscillator DMs featured in this Table. In particular, it is seen from FOM in Table 3 that the characterizations of both a-Si films using TLUR are less accurate than those via TLUF. The main reason for this is that TLUR employs the formula *ε*_i_(*E*) = const × *E* × exp(*E*/*E*_U_) for the film in the entire range *E* ≤ *E*_b_ [29], which, however, is inaccurate in the region of weak absorption from the measured UV/Vis/NIR spectrum *R*(*λ*). Instead, in this region *ε*_i_(*E*) = 2*n*(*E*)*k*(*E*) ~ *k*(*E*) = *αλ*/(4π) = 1239.8*α*/(4π*E*) ≅ const/*E* × exp(*E*/*E*_U_), taking into account the expression about *α* from Equation (3) and assuming the existence of Urbach tail in this region represented by Equation (2), whereas exactly the last formula for *ε*_i_(*E*) is used in TLUF [27].

It was indicated in the introduction that *n*_c_(*λ*) and *k*_c_(*λ*) are expressed in the framework of every broad-spectrum electron oscillator DM as functions only of the oscillator parameters. Correspondingly, the computed reflectance spectrum *R*_c_(*λ*) for such DM is obtained via the substitution of the computed film thickness parameters $\bar{d}$_c_ and ∆*d*_c_ and the oscillators parameters in Equation (4). Therefore, a larger absolute value of the difference *R*_c_(*λ*) − *R*(*λ*) in a particular spectral region indicates a larger inaccuracy of its respective DM in this region. With regard to this, the differences *R*_c_(*λ*) − *R*(*λ*) for the single oscillator DMs 1TL, 1NA, TLUR, and TLUF are computed, using the data from Table 3, and illustrated in Figure 8a,b, for both a-Si films.

### 3.4. Characterizations of the Films via Multiple Electron Oscillator DMs

Aiming to resolve the above-mentioned problem of the insufficient accuracy of characterization via a single oscillator DM in the region of medium absorption, both films are also characterized based on using three electron oscillator DMs 3TL and 3UD. To additionally reduce *FOM*, two single electron oscillators, 1PE and 1GA, are added to 3TL, as in [25], whereas the initial approximations of the central energies of 1PE and 1GA are above and below those of 3TL, respectively; the resultant DM being abbreviated as “3TL with 1PE and 1GA”. Moreover, Urbach tail and 1GA are added to 3UD, as in [31], and this DM is abbreviated to “3UD with UT and 1GA”. The results from the characterizations of the a-Si films via 3TL with 1PE and 1GA, as well as via 3UD with UT and 1GA are included in Table 4.

The a-Si films are also characterized via DMs, including two electron oscillators 2TL or 2UD as well as 1GA and 1PE or UT, whereas these DMs are abbreviated to “2TL with 1PE and 1GA” and “2UD with UT and 1GA” (they are simplifications of “3TL with 1PE and 1GA” and “3UD with UT and 1GA”). It turns out, though, that 2TL with 1PE and 1GA, as well as 2UD with UT and 1GA, are also insufficiently accurate in the region of medium absorption, for both a-Si films, similar to the single oscillator DMs 1TL, 1NA, TLUR, and TLUF, whose performance is featured in Table 3. Therefore, the results from the characterization of the a-Si films via 2TL with 1PE and 1GA, as well as via 2UD with UT and 1GA are not included in this paper.

### 3.5. Characterizations of the Films via a Synthetic DM including OEMR

This novel synthetic DM represents a constrained combination of OEMR for longer wavelengths and an existing electron oscillator DM for shorter wavelengths. It employs $\bar{d}$_c_ and ∆*d*_c_ computed at step B17 of the algorithm of OEMR, *n*_c_(*λ* ≥ *λ*_w_) is calculated from the Wemple–DiDomenico plot at step C2 of the algorithm, where *λ*_w_(nm) = 1239.8/*E*_w_(eV), and *k*_c_(*λ* ≥ *λ*_c_) is computed from the envelope *R*_+_(*λ*) at step C3 of the algorithm, where *λ*_c_(nm) = 1239.8/*E*_c_(eV). Both *n*_c_(*λ* ≤ *λ*_w_) and *k*_c_(*λ* ≤ *λ*_c_) are obtained via the electron oscillator DM, whereas their respective *n*_c_(*λ* ≤ *λ*_w_) and *k*_c_(*λ* ≤ *λ*_c_) are constrained to be identical with those for OEMR. The electron oscillator DM for shorter wavelengths, employed for the characterization of the as-Si films, is selected to be 3TL or 3NA (either of them containing three oscillators), thus having similarity to the already used 3TL with 1PE and 1GA as well as 3UD with UT and 1GA. The corresponding synthetic DMs are abbreviated as “OEMR with 3TL” and “OEMR with 3NA”. Computed parameters from characterizations of the a-Si films via OEMR with 3TL and OEMR with 3NA are also included in Table 4. Notably, in all four DMs featured in Table 4 are used the physically plausible approximations *n*_c_(∞) = 1 and *k*_c_(∞) = 0, as indicated, e.g., in [25,31].

A comparison of the FOM data from Table 3 and Table 4 shows that the smallest FOMs are achieved in the characterizations via the synthetic dispersion model OEMR with 3NA, for both films SP1 and SP2, thus indicating that these characterizations are the most accurate ones for these films. For this reason, the computed reflectance spectra *R*_c_ corresponding to the characterizations of the films SP1 and SP2 via OEMR with 3NA are included as dashed red lines in Figure 6c,d. Also, the fact that FOM is smaller for the characterization of either of the films SP1 and SP2 via OEMR with 3NA compared to OEMR with 3TL (as seen from Table 4) indicates that the appearances of *n*(*λ* ≤ *λ*_w_) and *k*(*λ* ≤ *λ*_c_) of these films can be explained using 3NA oscillators rather than 3TL oscillators.

Moreover, in Figure 8c,d are also included the differences *R*_c_(*λ*) − *R*(*λ*) corresponding to the characterizations of the a-Si films via the four DMs featured in Table 4. Furthermore, the extinction coefficient *k*(*λ*) of the film can be expressed via its absorption coefficient *α*(*E*), whereas *k*(*λ*) = *λα*(*λ*)/(4π) = 1239.8*α*(*E*)/(4π*E*) from Equation (3), and a logarithm from *α*(*E*) would be used for the representation of the Urbach tail according to Equation (2). With respect to the above, the refractive index *n*_c_(*λ*) and log10*α*_c_(*E*), computed using the four DMs featured in Table 4, are shown in Figure 9. Taking into account that the smallest FOMs in the characterizations of the a-Si films are achieved by employing OEMR with 3NA (see Table 3 and Table 4), the most accurate *n*_c_(*λ*) and log10*α*_c_(*E*) are shown in red colors in Figure 9.

A comparison between Figure 6a,c and Figure 9a,c, shows that the refractive index of the film SP1 is smaller than that of the film SP2, in the region of weak absorption. This is undoubtedly due to the presence of voids with a larger volume fraction *f*_void_ ≈ 5.4% for the film SP1 compared to *f*_void_ ≈ 0.4% for the film SP2, as described in the paragraph after Equation (17).

It is noticed from Figure 9a,c that *n*_c_(*λ*) corresponding to the characterizations of the a-Si films via OEMR with 3NA, and depicted by the red colored lines, is significantly larger in the region of strong absorption compared to the characterizations based on 3TA or 3UD in this region. This result is in agreement with the comment after Figure 1 that the refractive index *n*{*E* = [0,max(*ε*_r_)]} is larger for 1NA compared to 1TL, 1CC, and 1GA. Moreover, according to Equation (2), the insufficient linearity of the dependencies log10*α*_c_(*E*), presented in red color in Figure 9b,d, for photon energies *E* below *E*_g_ (taken from the data printed in red in Table 4) indicates that the assumption of Urbach tail is inaccurate, especially for the film SP2. A detailed analysis of the optical and electrical characteristics of such a-Si films, based on similar results for *n*_c_(*λ*) and log10*α*_c_(*E*), is intended to be published elsewhere.

## 4. Discussion

In the characterization of a thin film on a thick planar substrate only from *R*(*λ*) via a broad-spectrum DM, the film thickness parameters are computed together with the oscillator parameters by fitting the computed spectrum *R*_c_(*λ*) to *R*(*λ*), although DM represents an approximation of the true dispersion. In the case of interference spectrum *R*(*λ*), however, it is possible to achieve higher accuracy of the computed film thickness parameters $\bar{d}$_c_ and ∆*d*_c_ by employing EMR because it uses the interference fringes Equation (5) in addition to *R*(*λ*).

Furthermore, the proposed OEMR represents an improvement of EMR from [67,78]. The advantages of OEMR (compared to EMR) are due to the following: accounting for the finite size and the absorption in the substrate, the external smoothing of *R*(*λ*) to offset the partial coherence of light in the film, the slit width correction, the enhanced drawing of both envelopes based on [58,89], and the optimized computed thickness parameters $\bar{d}$_c_ and ∆*d*_c_ derived in an optimized interval [*λ*_t_(*l*_2c_), *λ*_t_(*l*_1c_)]. In fact, these advantages of OEMR compared to EMR are identical to the advantages of OEMT compared to EMT. With respect to this, the data and comments regarding Table 1 show that such OEMT provides a more accurate computed average film thickness $\bar{d}$_c_ than EMT or a limited version of OEMT, for a-Si, a-As_40_S_60_ and a-As_98_Te_2_ films. Therefore, the characterization of these films using OEMR should be more accurate than EMR.

Furthermore, *RE*($\bar{d}$_c_) is calculated from Equation (15) for an EMR characterization of amorphous As_33_S_67_ film published in Table 2 from [78] (Δd was taken into account there, unlike assuming Δd = 0 in [67]), whereas the result is *RE*($\bar{d}$_c_) ≈ 0.393%. This can be compared with *RE*($\bar{d}$_c_) ≈ 0.099% for the a-Si film SP1 and *RE*($\bar{d}$_c_) ≈ 0.172% for the a-Si film SP2, taken from Table 2 of this paper. These data indicate that employing OEMR leads to decreasing the error in the computation of the average film thickness $\bar{d}$ by more than 2.2 times in comparison with using EMR.

Table 5 includes data about FOM and the thickness parameters $\bar{d}$_c_ and ∆*d*_c_ of both a-Si films already computed via OEMR with 3NA, TLUF (being the best characterization featured in Table 3), and the best characterizations not employing OEMR from Table 4. Taking into account the content of the above three paragraphs, it is assumed tentatively that the parameters $\bar{d}$_c_ and ∆*d*_c_ computed via OEMR with 3NA (they are the same as those from OEMR) are identical with their respective true thickness parameters $\bar{d}$ and ∆d for each of the two a-Si films. Based on these data, we calculated the relative errors *RE*_c_($\bar{d}$_c_) = ($\bar{d}$_c_ − $\bar{d}$)/$\bar{d}$ and *RE*_c_(∆*d*_c_) = (∆*d*_c_ − ∆d)/∆d corresponding to the characterizations via TLUF and by the best characterizations not employing OEMR from Table 4, as these errors are also included in Table 5. Therefore, *RE*_c_($\bar{d}$_c_) and *RE*_c_(∆*d*_c_) represent the relative errors in the computation of $\bar{d}$ and ∆d via TLUF or the best characterizations not employing OEMR from Table 4.

The results from Table 5 indicate that smaller FOM for a particular film characterization only from UV/Vis/NIR *R*(*λ*) is associated with smaller errors in the computation of both $\bar{d}$ and ∆d, as well as that the errors in the computation of $\bar{d}$ are smaller than those for ∆d. Based on the above comments from this section, it is argued that the most accurate thickness parameters $\bar{d}$ and ∆d of a thin semiconductor or dielectric film on a thick planar substrate can be computed, from only one quasi-normal incidence UV/Vis/NIR *R*(*λ*), by using the proposed here OEMR. Therefore, OEMR should provide the most accurate *n*(*λ*) and *k*(*λ*) of such film in its region of weak to medium absorption. Notably, OEMR is applicable for any kind of thick planar substrate, independent of its degree of absorption.

Furthermore, utilizing the already computed via OEMR $\bar{d}$ and ∆d together with a given broad spectrum oscillator DM should result in the computation of more accurate parameters of its oscillators, which should lead to the computation of more accurate *n*(*λ*) and *k*(*λ*) of the film. Aiming to inculcate this idea into DM, the proposed synthetic DM includes OEMR for longer wavelengths and an existing electron oscillator DM for shorter wavelengths. Therefore, it is not surprising that the most accurate characterizations of the a-Si films SP1 and SP2 over the entire measured spectrum are performed by employing such synthetic DM, which turns out to be OEMR with 3NA for these films.

It should be also possible to use OEMR for the characterization of optical metasurfaces, such as those reported in [99,100]. However, this would require development of a model of the reflection from the metasurface and a respective equation for its *R*(*λ*).

## Figures and Tables

**Figure 1 nanomaterials-13-02407-f001:**
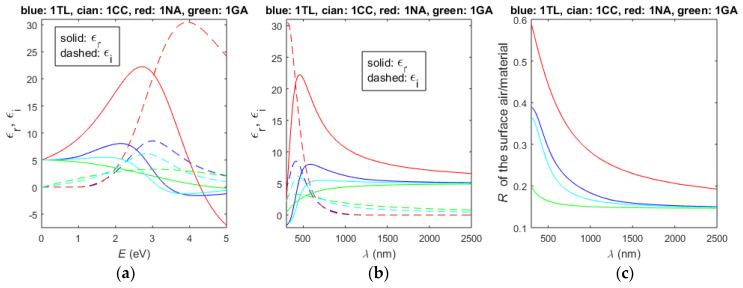
Calculated UV/Vis/NIR spectra for single oscillator DMs 1TL, 1CC, 1NA, and 1GA with identical central energy *E*_0_ = 3 eV, broadening term *B*_0_ = 2 eV, $\dot{\varepsilon }$(0) = $\dot{\varepsilon }$(*E* = 0) = *n*^2^(0) = 5 and $\dot{\varepsilon }$(∞) = $\dot{\varepsilon }$(*E*→∞) = *n*^2^(∞) = 1, as these values are selected to represent a semiconductor or dielectric material. (**a**) *ε*_r_(*E*) and *ε*_i_(*E*); (**b**) *ε*_r_(*λ*) and *ε*_i_(*λ*); (**c**) normal incidence *R*(*λ*) for the boundary air/material.

**Figure 2 nanomaterials-13-02407-f002:**
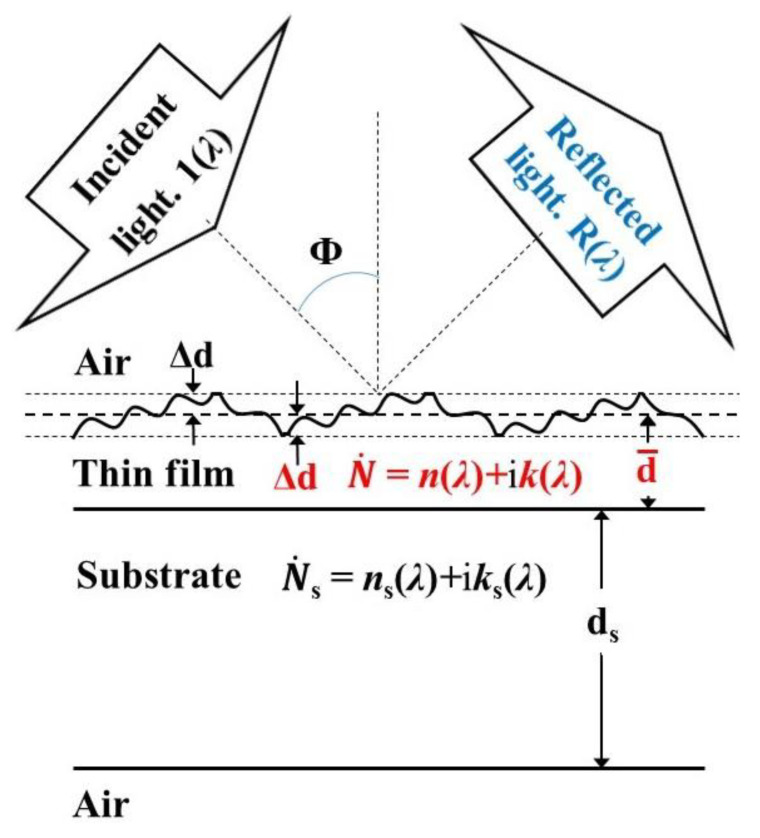
A sketch of reflectance *R*(*λ*) of a sample consisting of a non-uniform thin film on a thick planar substrate. The unknown optical characteristics of the film are printed in red color. The angle Φ is several degrees for quasi-normal incidence of light.

**Figure 3 nanomaterials-13-02407-f003:**
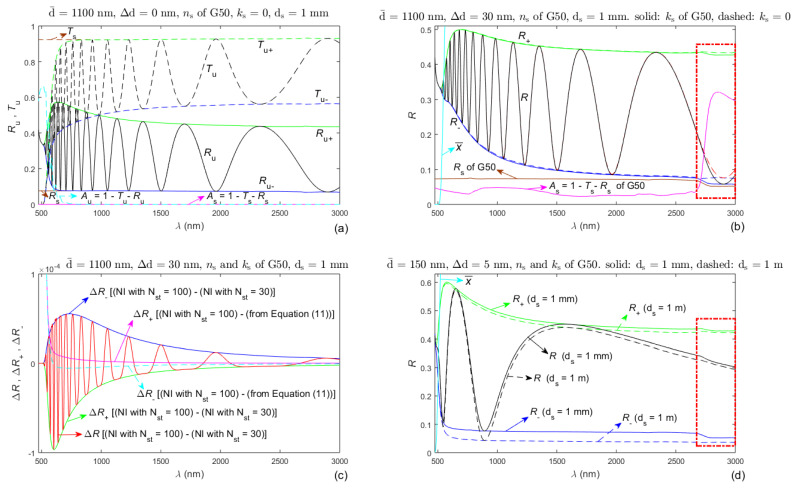
Simulated interference reflectance and transmittance spectra, their envelopes, and their features, for quasi-normal incidence of light to model specimens corresponding to a thin film a-Si:H on a thick planar glass substrate. (**a**) For a uniform film on a finite transparent substrate; (**b**) For two specimens of non-uniform film on finite transparent and non-transparent substrate, respectively; (**c**) For the non-uniform film on finite non-transparent substrate from (**b**); (**d**) For two specimens of much thinner non-uniform film on non-transparent finite and semi-infinite substrate, respectively. The region of significant absorbance in the substrate is exhibited by a dashed red colored rectangle in (**b**,**d**).

**Figure 4 nanomaterials-13-02407-f004:**
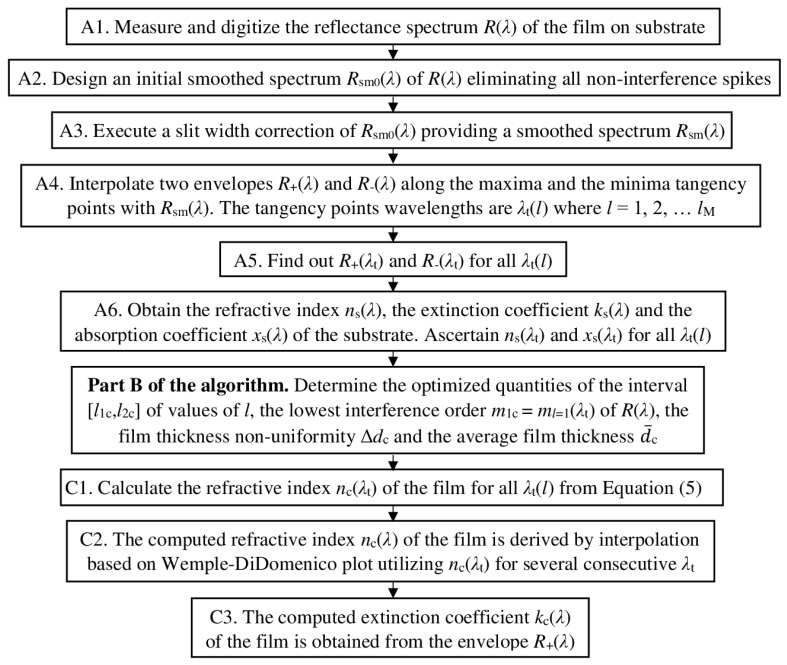
The algorithm of OEMR, exhibiting its detailed parts A and C for characterization of a thin semiconductor or dielectric film on a finite and generally non-transparent thick planar substrate.

**Figure 5 nanomaterials-13-02407-f005:**
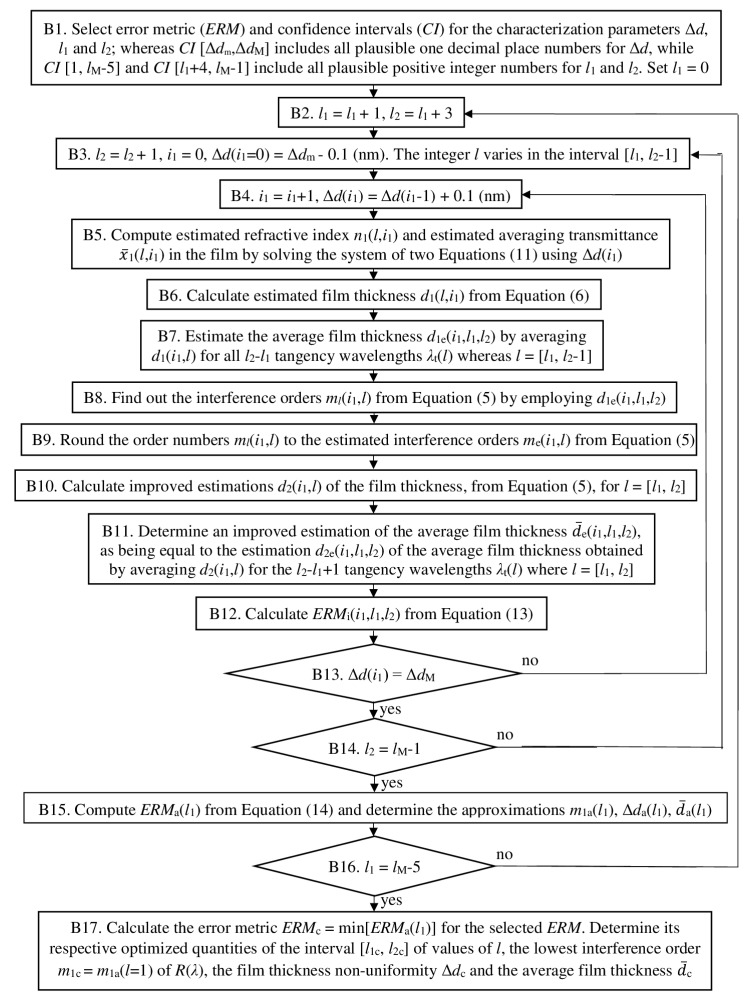
The optimizing part B of the algorithm of OEMR.

**Figure 6 nanomaterials-13-02407-f006:**
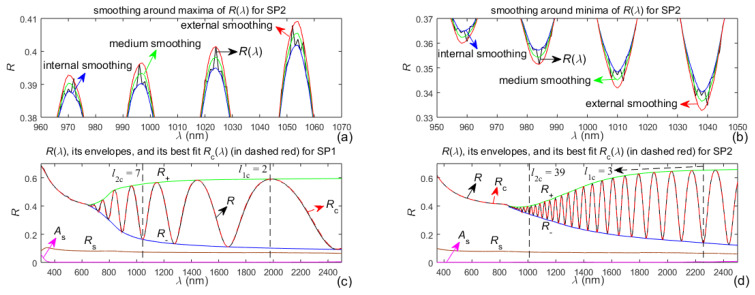
Reflectance spectra of the samples SP1 and SP2, their smoothing, and external envelopes. (**a**,**b**) Magnified images of the smoothing around some maxima and minima of *R*(*λ*) of SP2, respectively; (**c**) *R*(*λ*) of SP1, its external envelopes and substrate characteristics; (**d**) *R*(*λ*) of SP2, its external envelopes and substrate characteristics.

**Figure 7 nanomaterials-13-02407-f007:**
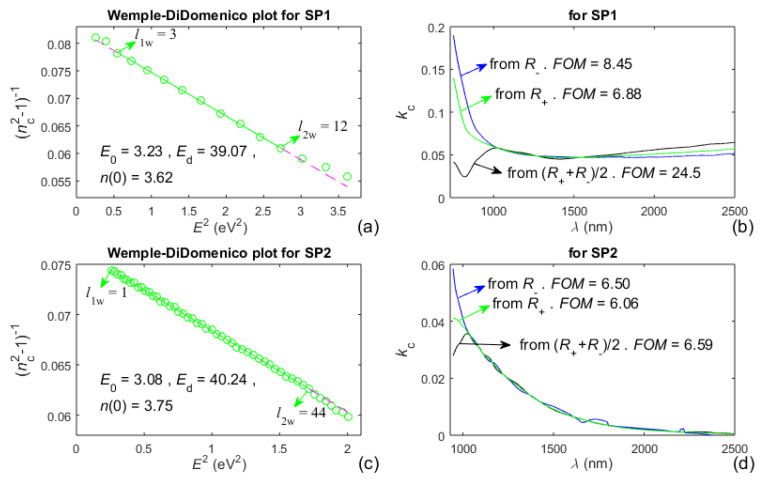
Results regarding the computation of *n*_c_(*λ*) and *k*_c_(*λ*) of both a-Si films via execution of part C from the OEMR algorithm. (**a**,**c**) Wemple-DiDomenico plots; (**b**,**d**) *k*_c_(*λ*) computed from one of the envelopes *R*_+_(*λ*), *R*_−_(*λ*), or their half-sum.

**Figure 8 nanomaterials-13-02407-f008:**
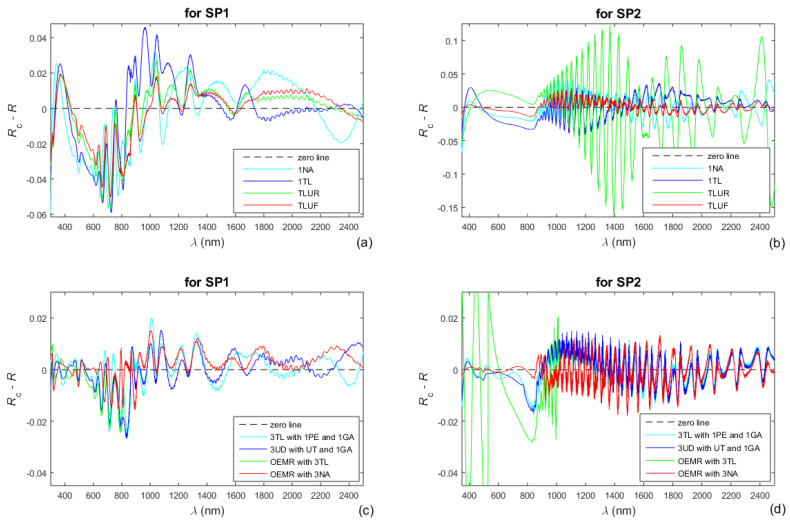
Differences *R*_c_(*λ*) − *R*(*λ*) between the computed and the measured reflectance spectra for several DMs (**a**,**b**) for the single oscillator DMs 1TL, 1NA, TLUR, and TLUF; (**c**,**d**) for the DMs featured in Table 4.

**Figure 9 nanomaterials-13-02407-f009:**
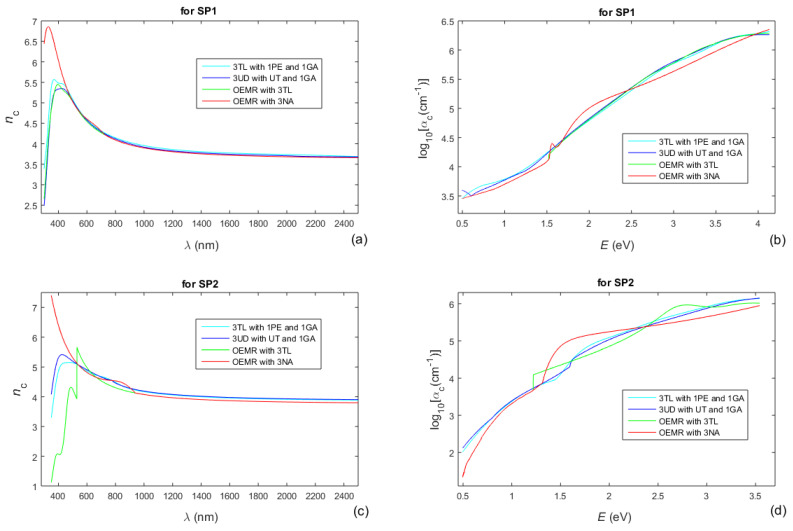
Spectral dependencies of *n*_c_ and log10*α*_c_ of the a-Si films SP1 and SP2, computed over the entire measured UV/Vis/NIR spectrum using each of the four DMs represented in Table 4. (**a**,**b**) For SP1; (**c**,**d**) For SP2. The most accurate *n*_c_(*λ*) and log10*α*_c_(*E*) are displayed by the red colored lines.

**Table 1 nanomaterials-13-02407-t001:** Computed data regarding the thickness parameters of amorphous thin films characterized via EMT or OEMT. The text and data in red color represent the most accurate results for these parameters.

Attributes of the Sample; and the Characterization	Film Material, Sample, [Reference]	$\bar{d}$_c_ (nm), Δ*d*_c_ (nm), [*l*_1c_, *l*_2c_]	RE($\bar{d}$_c_) (%)
uniform film, transparent substrate; **EMT**, common smoothing of *T*(*λ*), [*l*_1c_, *l*_2c_] is not optimized	a-Si, A029, [60]	1173, 0, [1, 12]	0.215
	a-Si, A074, [60]	1269, 0, [2, 14]	0.213
uniform film, absorbing substrate; **EMT**, common smoothing of *T*(*λ*), [*l*_1c_, *l*_2c_] is not optimized	a-As_40_S_60_, [56]	1409, 0, [11, 20]	0.261
non-uniform film, absorbing substrate; **EMT**, common smoothing of *T*(*λ*), Δ*d*_c_ and [*l*_1c_, *l*_2c_] are not optimized	a-As_40_S_60_, [56]	2689, 50, [3, 24]	0.496
non-uniform film, absorbing substrate; **OEMT**, external smoothing of *T*(*λ*), *l*_1c_ = 1, Δ*d*_c_ and *l*_2c_ are optimized	a-Si, A041, [58]	3929.9, 53.5, [1, 17]	0.245
	a-As_98_Te_2_, [59]	1983.8, 22.7, [1, 12]	0.133
non-uniform film, absorbing substrate; **OEMT**, external smoothing of *T*(*λ*), Δ*d*_c_ and [*l*_1c_, *l*_2c_] are optimized	a-Si, A041, [59]	3949.2, 53.0, [5, 14]	0.090
	a-As_98_Te_2_, [61]	1983.2, 23.9, [3, 9]	0.043

**Table 2 nanomaterials-13-02407-t002:** Results for the two a-Si films, computed by parts A and B from the algorithm of OEMR. The data corresponding to smallest *RE*($\bar{d}$_c_), for each of these films, are printed in red color, as their respective optimized thickness parameters $\bar{d}$_c_ and ∆*d*_c_ are in bold red.

FILM SP1						
*ERM*	*ERM* _c_	[*l*_1c_, *l*_2c_]	*m* _1c_	∆*d*_c_ (nm)	$\bar{d}$_c_ (nm)	*RE*($\bar{d}$_c_) (%)
DECONVOLUTION ENVELOPES OF *R*(*λ*)						
*ERM* _1_	6.61 × 10^−4^	[2, 7]	2	10.7	672.5	
*ERM* _2_	0.1263 nm	[2, 7]	2	10.7	672.5	0.1127
INTERNAL ENVELOPES OF *R*(*λ*)						
*ERM* _1_	5.63 × 10^−4^	[2, 7]	2	11.7	674.5	
*ERM* _2_	0.1144 nm	[2, 7]	2	11.8	674.7	0.1017
MEDIUM ENVELOPES OF *R*(*λ*)						
*ERM* _1_	5.64 × 10^−4^	[2, 7]	2	11.6	674.5	
*ERM* _2_	0.1127 nm	[2, 7]	2	11.7	674.5	0.1003
EXTERNAL ENVELOPES OF *R*(*λ*)						
* ERM * _ 1 _	4.62 × 10^−4^	[2, 7]	2	11.4	674.2	
* ERM * _ 2 _	0.1115 nm	[2, 7]	2	11.5	674.4	0.0992
**Optimized thickness parameters**				**11.5**	**674.3**	
**FILM SP2**						
*ERM*	*ERM* _c_	[*l*_1c_, *l*_2c_]	*m* _1c_	∆*d*_c_ (nm)	$\bar{d}$_c_ (nm)	*RE*($\bar{d}$_c_) (%)
DECONVOLUTION ENVELOPES OF *R*(*λ*)						
*ERM* _1_	1.72 × 10^−3^	[7, 44]	10	42.5	3589.9	
*ERM* _2_	0.2540 nm	[5, 44]	11	39.8	3701.6	0.2745
INTERNAL ENVELOPES OF *R*(*λ*)						
*ERM* _1_	1.20 × 10^−3^	[8, 40]	11	40.7	3713.0	
*ERM* _2_	0.2100 nm	[5, 41]	11	40.5	3704.1	0.2098
MEDIUM ENVELOPES OF *R*(*λ*)						
*ERM* _1_	1.15 × 10^−3^	[4, 40]	12	37.5	3837.7	
*ERM* _2_	0.2133 nm	[4, 40]	12	37.4	3835.5	0.2224
EXTERNAL ENVELOPES OF *R*(*λ*)						
* ERM * _ 1 _	1.02 × 10^−3^	[1, 39]	12	37.0	3848.7	
* ERM * _ 2 _	0.1782 nm	[3, 39]	12	36.9	3845.4	0.1715
**Optimized thickness parameters**				**37.0**	**3847.1**	

**Table 3 nanomaterials-13-02407-t003:** Computed parameters from utilization of four broad spectrum single electron oscillator DMs, employed independently from each other for characterizations of the a-Si films SP1 and SP2. The results regarding the most accurate amongst the respective four characterizations are printed in blue color, for each of these films.

FILM SP1								
1TL								
*A*_TL1_ (eV)	*E*_TL1_ (eV)	*B*_TL1_ (eV)	*E*_g_ (eV)	*ε*_r_ (∞)	$\bar{d}$_c_ (nm)	Δ*d*_c_ (nm)	*FOM*	
91.25	3.281	1.064	1.256	2.152	742.8	30.6	19.16	
1NA								
*f* _NA1_	*E*_NA1_ (eV)	*B*_NA1_ (eV)	*E*_g_ (eV)	*n*_c_ (∞)	$\bar{d}$_c_ (nm)	Δ*d*_c_ (nm)	*FOM*	
0.1222	3.543	0.6026	0.000	2.604	712.2	25.7	19.30	
TLUR								
*A*_TL1_ (eV)	*E*_TL1_ (eV)	*B*_TL1_ (eV)	*E*_g_ (eV)	*E*_b_ (eV)	$\bar{d}$_c_ (nm)	Δ*d*_c_ (nm)	*FOM*	
134.0	3.313	1.269	1.450	2.236	729.9	27.9	14.65	
TLUF								
* A * _ TL1 _ (eV)	* E * _ TL1 _ (eV)	* B * _ TL1 _ (eV)	* E * _ g _ (eV)	* E * _ b _ (eV)	* ε * _ r _ ( ∞ )	$\bar{d}$ _ c _ (nm)	Δ*d*_c_ (nm)	* FOM *
130.2	3.326	1.304	1.394	2.211	1.000	718.3	24.8	13.66
**FILM SP2**								
1TL								
*A*_TL1_ (eV)	*E*_TL1_ (eV)	*B*_TL1_ (eV)	*E*_g_ (eV)	*ε*_r_ (∞)	$\bar{d}$_c_ (nm)	Δ*d*_c_ (nm)	*FOM*	
46.61	2.990	0.9331	0.6684	5.992	3846.0	40.0	17.35	
1NA								
*f* _NA1_	*E*_NA1_ (eV)	*B*_NA1_ (eV)	*E*_g_ (eV)	*n*_c_ (∞)	$\bar{d}$_c_ (nm)	Δ*d*_c_ (nm)	*FOM*	
0.1151	3.224	0.5885	0.000	3.197	3700.5	35.9	15.78	
TLUR								
*A*_TL1_ (eV)	*E*_TL1_ (eV)	*B*_TL1_ (eV)	*E*_g_ (eV)	*E*_b_ (eV)	$\bar{d}$_c_ (nm)	Δ*d*_c_ (nm)	*FOM*	
88.81	2.694	2.177	0.6941	0.8318	3571.7	39.7	51.1	
TLUF								
* A * _ TL1 _ (eV)	* E * _ TL1 _ (eV)	* B * _ TL1 _ (eV)	* E * _ g _ (eV)	* E * _ b _ (eV)	* ε * _ r _ (∞)	$\bar{d}$ _ c _ (nm)	Δ*d*_c_ (nm)	* FOM *
96.78	3.382	1.550	0.8953	1.358	2.814	3701.9	33.5	7.65

**Table 4 nanomaterials-13-02407-t004:** Computed parameters from characterizations of the a-Si films based on four broad spectrum oscillator DMs, each of them including three oscillators from the same type in the spectral region of strong absorption in the film. The data regarding the most accurate characterizations of the films by these DMs are in red color.

FILM SP1									
3TL with 1PE and 1GA									
*A*_TL1_ (eV)	*E*_TL1_ (eV)	*B*_TL1_ (eV)	*A*_TL2_ (eV)	*E*_TL2_ (eV)	*B*_TL2_ (eV)	*A*_TL3_ (eV)	*E*_TL3_ (eV)	*B*_TL3_ (eV)	*E*_g_ (eV)
10.63	4.195	0.5072	57.52	3.673	0.9566	15.51	2.992	0.9086	0.7942
*A*_PE1_ (eV^2^)	*E*_PE1_ (eV)	*f* _GA1_	*E*_GA1_ (eV)	*B*_GA1_ (eV)	$\bar{d}$_c_ (nm)	Δ*d*_c_ (nm)	*FOM*		
0.0733	0.3493	0.3619	0.7070	0.4239	664.7	0.0664	6.80		
3UD with UT and 1GA									
*A* _UD1_	*E*_UD1_ (eV)	*B*_UD1_ (eV)	*A* _UD2_	*E*_UD2_ (eV)	*B*_UD2_ (eV)	*A* _UD3_	*E*_UD3_ (eV)	*B*_UD3_ (eV)	*E*_g_ (eV)
8.407	5.024	0.004290	92.22	3.600	0.5679	18.06	2.919	0.5170	1.209
*N* _vc_	*E*_h_ (eV)	*f* _UT_	*E*_U_ (eV)	*f* _GA1_	*E*_GA1_ (eV)	*B*_GA1_ (eV)	$\bar{d}$_c_ (nm)	Δ*d*_c_ (nm)	*FOM*
286.4	33.11	0.04288	0.7972	0.00055	0.00042	0.3111	670.4	5.0	6.64
OEMR with 3TL									
*A*_TL1_ (eV)	*E*_TL1_ (eV)	*B*_TL1_ (eV)	*A*_TL2_ (eV)	*E*_TL2_ (eV)	*B*_TL2_ (eV)	*A*_TL3_ (eV)	*E*_TL3_ (eV)	*B*_TL3_ (eV)	*E*_g_ (eV)
65.74	3.746	1.199	12.73	3.284	0.7021	8.399	2.790	0.7875	0.8749
*E*_w_ (eV)	*E*_c_ (eV)	*FOM*							
1.663	1.520	6.55							
OEMR with 3NA									
* f * _ NA1 _	* E * _ NA1 _ (eV)	* B * _ NA1 _ (eV)	* f * _ NA2 _	* E * _ NA2 _ (eV)	* B * _ NA2 _ (eV)	* f * _ NA3 _	* E * _ NA3 _ (eV)	* B * _ NA3 _ (eV)	* E * _ g _ (eV)
1.147	4.268	1.309	0.4397	1.686	0.4883	0.01164	1.545	0.03718	1.428
* E * _ w _ (eV)	* E * _ c _ (eV)	* FOM *							
1.718	1.528	5.46							
**FILM SP2**									
3TL with 1PE and 1GA									
*A*_TL1_ (eV)	*E*_TL1_ (eV)	*B*_TL1_ (eV)	*A*_TL2_ (eV)	*E*_TL2_ (eV)	*B*_TL2_ (eV)	*A*_TL3_ (eV)	*E*_TL3_ (eV)	*B*_TL3_ (eV)	*E*_g_ (eV)
99.56	3.236	1.158	32.91	2.544	0.9799	48.17	1.683	0.7690	1.424
*A*_PE1_ (eV^2^)	*E*_PE1_ (eV)	*f* _GA1_	*E*_GA1_ (eV)	*B*_GA1_ (eV)	$\bar{d}$_c_ (nm)	Δ*d*_c_ (nm)	*FOM*		
272.0	10.09	1.038	1.643	0.4271	3758.3	35.9	5.49		
3UD with UT and 1GA									
*A* _UD1_	*E*_UD1_ (eV)	*B*_UD1_ (eV)	*A* _UD2_	*E*_UD2_ (eV)	*B*_UD2_ (eV)	*A* _UD3_	*E*_UD3_ (eV)	*B*_UD3_ (eV)	*E*_g_ (eV)
14.56	3.271	0.6323	8.882	1.991	1.388	0.2774	1.5848	0.001321	1.593
*N* _vc_	*E*_h_ (eV)	*f* _UT_	*E*_U_ (eV)	*f* _GA1_	*E*_GA1_ (eV)	*B*_GA1_ (eV)	$\bar{d}$_c_ (nm)	Δ*d*_c_ (nm)	*FOM*
345.7	41.70	0.3301	0.1662	0.000716	1.2989	0.374692	3741.5	35.2	5.84
OEMR with 3TL									
*A*_TL1_ (eV)	*E*_TL1_ (eV)	*B*_TL1_ (eV)	*A*_TL2_ (eV)	*E*_TL2_ (eV)	*B*_TL2_ (eV)	*A*_TL3_ (eV)	*E*_TL3_ (eV)	*B*_TL3_ (eV)	*E*_g_ (eV)
5.579	3.300	0.6385	0.2499	2.837	0.3954	8.623	2.680	0.4194	0.00022
*E*_w_ (eV)	*E*_c_ (eV)	*FOM*							
2.338	1.219	16.22							
OEMR with 3NA									
* f * _ NA1 _	* E * _ NA1 _ (eV)	* B * _ NA1 _ (eV)	* f * _ NA2 _	* E * _ NA2 _ (eV)	* B * _ NA2 _ (eV)	* f * _ NA3 _	* E * _ NA3 _ (eV)	* B * _ NA3 _ (eV)	* E * _ g _ (eV)
0.7039	4.901	0.9377	0.01762	1.535	0.1140	0.8674	1.242	0.3389	1.232
* E * _ w _ (eV)	* E * _ c _ (eV)	* FOM *							
1.320	1.305	5.18							

**Table 5 nanomaterials-13-02407-t005:** *FOM*s and thickness parameters already computed using selected characterizations of the a-Si films, as well as relative errors in the computations of these parameters with respect to their values computed by OEMR.

FILM SP1					
Characterization method	*FOM*	$\bar{d}$_c_ (nm)	*RE*_c_($\bar{d}$_c_) (%)	∆*d*_c_ (nm)	*RE*_c_(∆*d*_c_) (%)
OEMR with 3NA	5.46	674.3	-	11.5	-
TLUF	13.66	718.3	6.5	24.8	115.7
3UD with UT and 1GA	6.64	670.4	−0.6	5.0	−56.5
**FILM SP2**					
Characterization method	*FOM*	$\bar{d}$_c_ (nm)	*RE*($\bar{d}$_c_) (%)	∆*d*_c_ (nm)	*RE*_c_(∆*d*_c_) (%)
OEMR with 3NA	5.18	3847.1	-	37.0	-
TLUF	7.65	3701.9	−3.8	33.5	−9.5
3TL with 1PE and 1GA	5.49	3758.3	−2.3	35.9	−3.0

## Data Availability

Data are available upon request.

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
