# Peer review of "Increasing the Accuracy of the Characterization of a Thin Semiconductor or Dielectric Film on a Substrate from Only One Quasi-Normal Incidence UV/Vis/NIR Reflectance Spectrum of the Sample"

_nanomaterials, 2023, doi:10.3390/nano13172407_

Round 1

Reviewer 1 Report

The manuscript titled by “Increasing the accuracy of characterization of a thin semiconductor or dielectric film on a substrate only from one quasi-normal incidence UV/Vis/NIR reflectance spectrum of the sample”, report an optimizing envelope method, named OEMR, for characterization of thin dielectric or semiconductor film using only one quasi-normal incidence UV/Vis/NIR reflectance spectrum R(λ) of the film on substrate. The method OEMR is developed for the common case of quasi-normal incidence of light to a specimen consisting of a thin semiconductor or dielectric film on a thick substrate with n(λ) > ns(λ) > 1. OEMR represents an improvement of EMR method, by accounting for the absorption in the substrate, external smoothing of R(λ) to offset the partial coherence of light in the film, slit width correction, enhanced drawing of both envelopes based, and optimized computed thickness parameters ?Ì…c and Δdc of the film. The authors use two a-Si films to show that employing OEMR can lead to superior accuracy characterization of semiconductor and dielectric films, only from R(λ), over the entire UV/Vis/NIR R(λ). In addition, the authors also show the characterization of both films based on three electron oscillator DMs, which reduced FOM comparing the characterization based on single oscillator DM.

The characterization of the thin films is an important topic in material science. The paper illustrates an optimized method to obtain the parameters of the films, such as thickness, refractive index and etc. based on reflectance spectrum only. The method OEMR is described in detail in this work, that is very helpful for the researchers in this field. Therefore, I recommend the publication of the paper after the authors clarify some minor issues in the paper.  

1. Eq. (3), line 265-270, the symbols used in Equation are very confused and not clearly descripted.

2.  The requirement of the substrate is unclear. What is the definition of non-transparent substrate? If the substrate is transparent, is the method OEMR applicable?

3.    Could the authors discuss the general application of the method OEMR?

Author Response

Remark 1: Eq. (3), line 265-270, the symbols used in Equation are very confused and not clearly descripted.                                                                                                                 Reply 1: Indeed, some symbols from Equation (3) are not defined explicitly. However, e.g. if the complex refractive indices  of the film and  of the substrate are known, then e.g. the complex number   is calculated from the expression whereas  is the amplitude of   and is its phase. Since our software includes the command “Re” and “Im” for real part and imaginary part of a complex number, the above expression is convenient for computation of  and utilizing complex numbers. Correspondingly, the following sentence is added after Equation (3): “The expression of Ru(λ) in Equation (3) by using complex numbers is convenient for development of computer code for optical characterization of thin films.”

Remark 2: The requirement of the substrate is unclear. What is the definition of non-transparent substrate? If the substrate is transparent, is the method OEMR applicable?              

Reply 2: The implied meaning of non-transparent substrate is that it has ks(λ ) > 0 within some wavelengths region from the measured spectrum. Therefore, the following text is added within the first sentence from subsection 2.1.: “(with ks > 0 for some λ)“. Furthermore, OEMR is applicable for transparent, absorbing or opaque substrate, i.e. for any kind of thick planar substrate. Correspondingly, the following sentence is added at the end of the fifth paragraph from section 4: “Notably, OEMR is applicable for any kind of thick planar substrate, independent of its degree of absorption.”

Remark 3: Could the authors discuss the general application of the method OEMR?                      

Reply 3: In the paragraph after Table 5 from our original manuscript is written: “…it is argued that most accurate thickness parameters  and ∆d of a thin semiconductor or dielectric film on a thick planar substrate can be computed, from only one quasi-normal incidence UV/Vis/NIR R(λ), by using the proposed here OEMR.“. Immediately after this sentence in the corrected manuscript is included the sentence “Based on this, OEMR should provide most accurate n(λ) and k(λ) of such film in its region of weak to medium absorption.”.   

Besides, the text in the next paragraph explains our belief that the proposed here synthetic DM including OEMR for longer wavelengths and an existing electron oscillator DM for shorter wavelengths should provide the most accurate n(λ) and k(λ) of such film from only one quasi-normal incidence UV/Vis/NIR R(λ).

Reviewer 2 Report

Minkov and coworkers discuss the exclusive use of quasi normal reflectance spectroscopy for the characterization of thin films (thickness, refractive index, …) developing a method with some elements of novelty with respect to a state-of-art, conventional method. Si thin films are used as validation samples

The manuscript is well organized and sufficiently clear

It extensively reports technicalities of the work –in the theory, model and experiments - thus enabling the check of the reported results 

It is not very clear to which extent the work  brings novelty to the field and provide to the community a new tool that can be applied to virtually any thin film or layered structure

It is not clear how and if this work may help the study and engineering of  systems such as metamaterials, photonic crystals, metasurfaces (e.g. optical or plasmonic ones).

Without this strong statement, the work could be interpreted as a style exercise, not very useful…

the authors must take the effort to widen the discussion and /or application of their model to thin film structures more complex and useful than a silicon layer, for instance by choosing additional cases of study. 

Contextually, the discussion of the state or the art should be widened to include different families of nanostructured surfaces.

The work should take this new, more exciting flavor  in order to be considered for publication in Nanomaterials.

Technical notes

OEMT: the first work in the abstract, clearly an acronyms, never defined….. 

Referee suggestion for widening the class of nanomaterials discussed and referred to in the manuscript: 

- plasmonic metasurfaces (e.g. ACS Photonics 2, 675 - 67917 , 2015)

- optical metasurfaces (e.g. Nanomaterials  7, 2017,  400 and also Materials, 2019, 12, 3572)

Author Response

Remark 1: It is not very clear to which extent the work brings novelty to the field and provide to the community a new tool that can be applied to virtually any thin film or layered structure. Reply 1: The principle novelty of OEMR is summarized in the first two paragraphs from Section 4. In general, OEMR provides most accurate thickness parameters and ∆d of the film, as well as n(λ) and k(λ) in the region of weak to medium absorption, compared to all existing methods using only one measured UV/VIS/NIR reflectance spectrum R(λ). The principle novelty of the synthetic DM including OEMR is summarized in the penultimate paragraph from Section 4 of the corrected manuscript. In general, it uses  and ∆d already computed by OEMR and provides most accurate n(λ) and k(λ) over the entire measured UV/VIS/NIR R(λ), compared to all existing methods using only one such spectrum R(λ).     

Remark 2: It is not clear how and if this work may help the study and engineering of systems such as metamaterials, photonic crystals, metasurfaces (e.g. optical or plasmonic ones).

Without this strong statement, the work could be interpreted as a style exercise, not very useful.

Reply 2: The mechanism of reflection used in the derivation of Equation (3) from the proposed paper is illustrated in the left side of the below Figure, whereas it assumes existence of a relatively flat, although non-uniform surface air/film (see Figure 2 from the paper). This mechanism does not include reflectance from object perpendicular to the surface, such as that for the optical metasurface from Materials, 2019, 12, 3572, illustrated in the right side of the below Figure. Therefore, application of our approach to such a case should be possible, however it requires modification of Equation (3) to account for the geometry of the investigated system. Regarding the usefulness of our work, it is described in short in the above Reply 1.

Remark 3: The authors must take the effort to widen the discussion and /or application of their model to thin film structures more complex and useful than a silicon layer, for instance by choosing additional cases of study. 

Reply 3: The a-Si films SP1 and SP2 studied in our paper differ significantly in their thicknesses (as seen from Table 2 in our paper) and structure. To elucidate this, the following paragraph is included after Figure 9: “A comparison between Figures 9a, 9c, 6a and 6c shows that the refractive index of the film SP1 is smaller than that of the film SP2, in the region of weak absorption. This is undoubtedly due to the presence of voids with larger volume fraction fvoid ≈ 5.4% for the film SP1 compared to fvoid ≈ 0.4% for the film SP2, as described in the paragraph after Equation (17).”

Besides, in the third paragraph from Section 4 of our paper are presented data about As33S67 film, showing that employing OEMR leads to decreasing the error in computation of the average film thickness  by more than 2.2 times in comparison with using EMR. In general, using more accurate  results in more accurate overall characterization of the studied film, as indicated by the data from Table 5 and the simplified description from our Reply 1.

Furthermore, to expand our study to different materials, the following text is added at the end of the second paragraph from Section 4: “In fact, these advantages of OEMR compared to EMR are identical with the advantages of OEMT compared to EMT. With respect to this, the data and comments regarding Table 1 show that such OEMT provides more accurate computed average film thickness c than EMT or limited version of OEMT, for a-Si, a-As40S60 and a-As98Te2 films. Therefore, characterization of these films by using OEMR should be more accurate than by EMR.”      

Remark 4: Contextually, the discussion of the state or the art should be widened to include different families of nanostructured surfaces. The work should take this new, more exciting flavor in order to be considered for publication in Nanomaterials. Referee suggestion for widening the class of nanomaterials discussed and referred to in the manuscript: 

- plasmonic metasurfaces (e.g. ACS Photonics 2, 675 - 67917 , 2015)

- optical metasurfaces (e.g. Nanomaterials  7, 2017,  400 and also Materials, 2019, 12, 3572)

Reply 4: To address this issue, the following text is included at the end of Section 4: “It should be also possible to use OEMR for characterization of optical metasurfaces, such as those reported in [99,100], however this would require development of a model of the reflection from the metasurface and respective equation for its R(λ).” Correspondingly, the following Journal papers are added to the References:

  1. Floris, F.; Fornasari, L.; Marini, A.; Bellani, V.; Banfi, F.; Roddaro, S.; Ercolani, D.; Rocci, M.; Beltram, F.; Cecchini, M.; Sorba, L.; Rossella, F. Self-Assembled InAs Nanowires as Optical Reflectors. Nanomaterials 2017, 7, 7110400:1-11.
  2. Floris, F.; Fornasari, L.; Bellani, V.; Marini, A.; Banfi, F.; Marabelli, F.; Beltram, F.; Ercolani, D.; Battiato, S.; Sorba, L.; Rossella, F. Strong Modulations of Optical Reflectance in Tapered Core–Shell Nanowires, Materials 2019, 12, 12213572:1-11.

Moreover, the three papers mentioned in Remark 4 contain respectively 5 pages, 11 pages and 11 pages, while our proposed paper has 30 pages. Therefore, in our opinion, such characterization of optical metasurfaces should not be performed in this our paper.

Besides, using OEMR requires presence of at least three apparent maxima and three apparent minima of interference spectrum R(λ), as stated in the paragraph after Equation (4) from our paper. Since this is not the case for the plasmonic metasurfaces from ACS Photonics 2, 675 - 67917 , 2015, OEMR can not be used for characterization of these plasmonic metasurfaces.

Regarding publishability of our paper, in the scope of Nanomaterials of MDPI is written “Nanomaterials are materials with typical size features in the lower nanometer size range…”, and as examples are given: “coatings and thin films…”. With respect to this, in our paper is established that the thickness non-uniformity of the film SP1 is ∆d = 11.5 nm, and the effective thickness filled with voids is 15.4 nm for the film SP2.

Furthermore, before submitting our paper to Nanomaterials, we wrote Mr. Steve Yan, Assistant editor, Nanomaterials Editorial Office/MDPI with a request about the publishability of our paper, and in his reply dated 29.05 he has written “Your research holds significant potential, and we would be thrilled to have it featured in our publication.”

Remark 5: Technical notes. OEMT: the first work in the abstract, clearly an acronyms, never defined.

Reply 5: In the penultimate paragraph in page 5 of our original paper was written “Furthermore, we improved EMT by developing an optimizing envelope method for T(λ), abbreviated as OEMT [57-59].”. This text is not changed in the corrected manuscript.

Reviewer 3 Report

The manuscript is very well written and structured, and the described methods will be of interest to other readers of Nanomaterials.

In my opinion, the paper is acceptable for publication in the present form.

English is generally good. Minor corrections can be implemented during the proofs. As an example in some sentences, the subject is misplaced after the verb (see ln. 23, 103, 138)

Author Response

In the report of Reviewer 3 is written: “In my opinion, the paper is acceptable for publication in the present form.”. Therefore, we have not made any changes, originating from this report, in the manuscript.
